# On-chip mid-infrared photothermoelectric detectors for full-Stokes detection

Mingjin Dai[1], Chongwu Wang[1], Bo Qiang[1], Fakun Wang[1], Ming Ye[1], Song Han[1], Yu Luo [1] ✉ & Qi Jie Wang [1,2] ✉

On-chip polarimeters are highly desirable for the next-generation ultra-compact optical and optoelectronic systems. Polarization-sensitive photodetectors relying on anisotropic absorption of natural/artificial materials have emerged as a promising candidate for on-chip polarimeters owing to their filterless configurations. However, these photodetectors can only be applied for detection of either linearly or circularly polarized light, not applicable for full-Stokes detection. Here, we propose and demonstrate three-ports polarimeters comprising on-chip chiral plasmonic metamaterial-mediated mid-infrared photodetectors for full-Stokes detection. By manipulating the spatial distribution of chiral metamaterials, we could convert polarization-resolved absorptions to corresponding polarization-resolved photovoltages of three ports through the photothermoelectric effect. We utilize the developed polarimeter in an imaging demonstration showing reliable ability for polarization reconstruction. Our work provides an alternative strategy for developing polarization-resolved photodetectors with a bandgap-independent operation range in the mid-infrared.

State of polarization (SoP) characterizing the electric field oscillation is essential for optic-related applications such as optical communication, remote sensing, and navigation[1–3]. Mid-infrared (mid-IR) polarization detectors are especially attractive owing to their widespread applications in chemical analysis, biomedical diagnosis, and face recognition[4–6]. For decades, conventional polarization detection approaches include division-of-time, division-of-amplitude, division-of-aperture, and division-of-focal-plane, which normally requires a combination of linear retarders, polarizers, half-wave plates, and quarter-wave plates[7,8]. However, such bulky and complicated optical systems by using free-space polarizer have intrinsic drawbacks such as limited speed, limited accuracy, and incomplete polarization state detection[9]. Recent advances in low-dimensional nanophotonic technologies have unveiled fascinating approaches to develop the next-generation polarimeters[10,11]. As a potential candidate for the next-generation compact polarimeters, on-chip polarization-sensitive photodetectors have been widely studied recently owing to their

advantages including high level of miniaturization and ultrahigh-density integration.

Up to date, one of the main approaches to detect SoP is based on the structural anisotropy or chirality from the natural materials. In general, photodetectors for linear polarization detection rely on the anisotropic absorption of one-dimensional nanowires or two-dimensional van der Waals materials[12–14], while photodetectors for circular polarization detection are based on the chiral absorption of light in organic semiconductors and hybrid perovskites[15,16], the spin photogalvanic effect in topological insulator or semimetals[17–20], inverse spin Hall effect at metal-semiconductor interface[21,22], and spin-dependent recombination of conduction electrons[23,24]. However, the applications of these polarization-sensitive photodetectors are hindered by intrinsic limitations, such as bandgap-dependent spectral responses, chemical instability, and low polarization sensitivity associated with small anisotropy or chirality. In addition, most of these polarization-sensitive photodetectors only work for detection of either

[1]School of Electrical and Electronic Engineering, Nanyang Technological University, Singapore 639798, Singapore. [2]Centre for Disruptive Photonic Technologies, School of Physical and Mathematical Sciences, Nanyang Technological University, Singapore 637371, Singapore. ✉e-mail: luoyu@ntu.edu.sg; qjwang@ntu.edu.sg

linear polarization or circular polarization of light but cannot be applied to full-Stokes detection. Because artificial structures can achieve strong anisotropy and chirality, and have a great design flexibility and a filterless configuration, such functional photodetectors enabled by artificial structures can realize compact polarimetry for polarized light detection, as well as polarization imaging with potentially ultra-high pixel density. Using artificial structures integrated with active materials is another main approach for detection of SoP. This

approach led to polarization-sensitive photodetectors operating in scattered, absorbed, and guided radiation modes[25]. As an example, plasmonic metamaterials with polarization-selective field enhancements have been integrated with semiconductors to generate polarization-sensitive photocurrents[26,27]. However, most of previous detectors relied on the photoconductive or the photovoltaic effect, which requires matching between the resonant wavelength of plasmonic metamaterials and the bandgap of semiconductors[7,28].

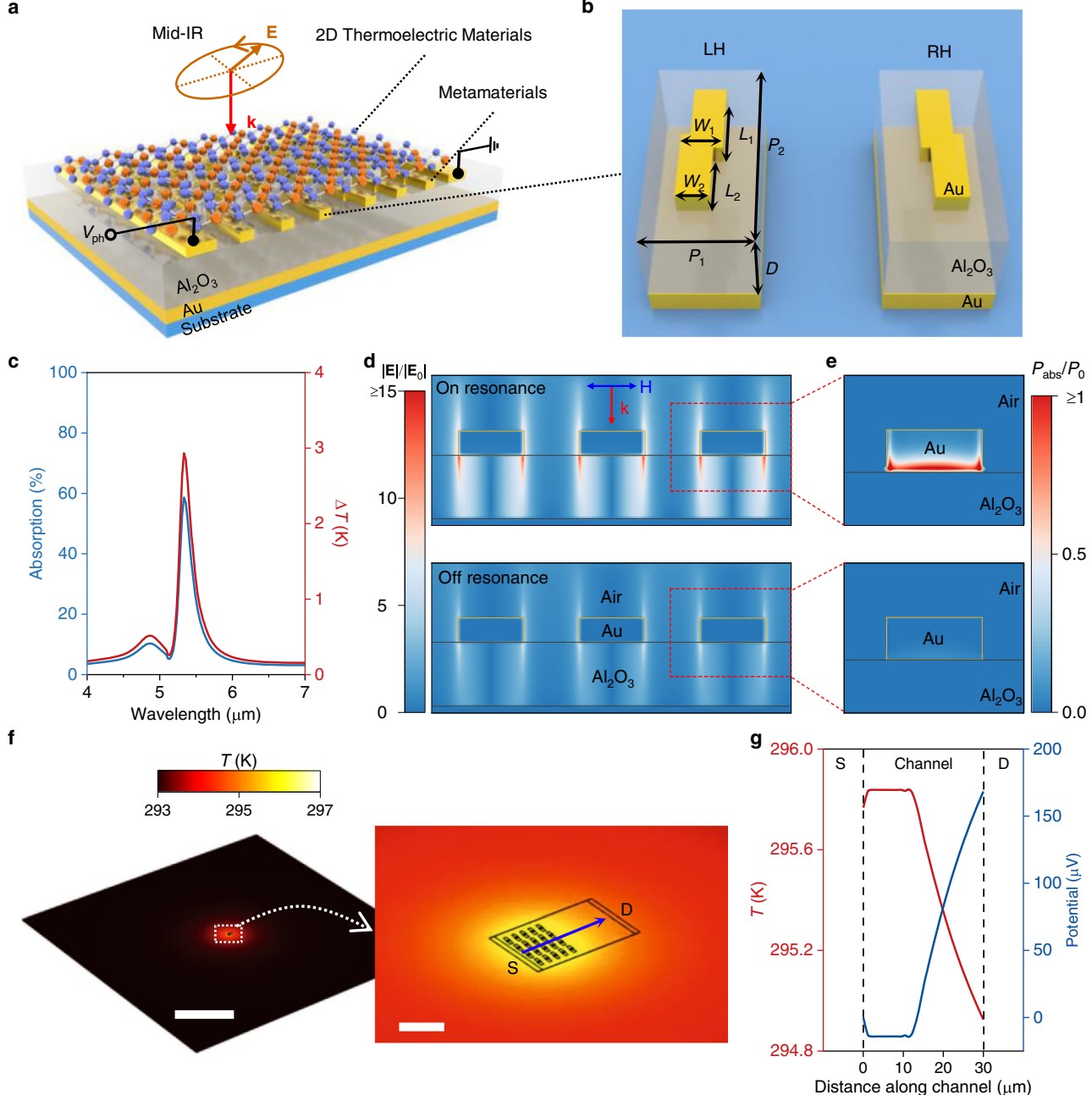

**Fig. 1 | Design principle of mid-IR photodetector based on resonance photothermoelectric response. a** Device architecture design for resonance thermoelectric photoresponse. Here, **k**, **E**, and $V_{ph}$ represent wavevector, electric field vector, and photovoltage, respectively. **b** Schematic of the chiral metamaterial consisting of the chiral plasmonic meta-molecule array (Au), dielectric spacer ($Al_2O_3$), and metal backplane (Au). LH and RH denote left-handed and right-handed, respectively. $W_{1,2}$, $L_{1,2}$, $P_{1,2}$, and $D$ indicate the width, length, periodic scale, and thickness of $Al_2O_3$, respectively. **c** Simulated absorption, and temperature increase ($\Delta T$) as a function of wavelength with input power of

5 mW. **d**, **e** Full wave simulations of electric field distributions (**d**) normalized to incident electric field, and power absorption density (**e**) is calculated by $P_{abs} = 1/2\omega\varepsilon''|\mathbf{E}|^2$, and is normalized by $P_0$, the incident power divided by the meta-molecule volume. **f** A thermal simulation of a typical device with half-side chiral metamaterial at peak absorption with input power of 5 mW. S and D denote source and drain electrodes, respectively. Scale bars, 400 μm (left: main image); 10 μm (right: inset). **g** The corresponding temperature and potential profiles across the device channel. Two vertical dash lines indicate the interfaces between electrodes and channel.

Therefore, an efficient way to transfer strong anisotropy and chirality to electrical readouts without the operation wavelength limitation by the bandgap of active materials is highly desired.

Standard SoP detection requires four measurements to obtain four Stokes parameters, namely the light intensity, two linear polarization components, and one circular polarization component[29–31]. In other words, both linear and circular polarization-sensitive photodetectors are needed for the full-Stokes detection. For linear and circular polarization-sensitive photodetectors, the crucial figure of merits to characterize the polarization sensitivity are the polarization ratio (PR) and dissymmetry factor ($g$). The large PR and $g$ values are highly critical for improving the accuracy of detection in practical applications. However, most previously reported polarization-sensitive photodetectors usually exhibit unipolar polarization-dependent photoresponses and the corresponding PR and $g$-factor are generally small, e.g., $0 < PR < 20$ and $0 < g < 2$. It is noted that, the PR and $g$ factor are calculated using $PR = V_{max}/V_{min}$ and $g = 2 \times (V_{LCP} - V_{RCP})/(V_{LCP} + V_{RCP})$, where $V_{max}$ and $V_{min}$ represent the maximum and the minimum linear-polarization-dependent photovoltage, respectively, and $V_{LCP}$ and $V_{RCP}$ denote the photovoltages under left-handed circular polarized (LCP) and right-handed circular polarized (RCP) light illuminations, respectively[32]. To increase the PR value, the bipolar linear-polarization-sensitive photodetectors have been recently realized by introducing the Dember effect modulated by the photonic mechanism and the hot-carrier mechanism by integrating nanoantenna on graphene[14,33]. As a result, the PR can be controlled to have values in the range of $(1 \rightarrow \infty / -\infty \rightarrow -1)$ with a transition from positive to negative. However, such a realization only demonstrated the bipolar linear polarization detection, while the polarity transition for circular polarization detection has not been realized till today. On the other hand, the robust detection of circular polarized light with immunity against the ubiquitous unpolarized and linearly polarized light has not been realized.

Here, we aim to tackle such a challenge and achieve the bipolar linear and circular polarization detection, simultaneously, for the development of a monolithic full-Stokes polarimeter. Leveraging on the SoP-dependent photothermal effect in plasmonic chiral metamaterials and Seebeck effect in two-dimensional (2D) thermoelectric materials, we demonstrate experimentally room-temperature mid-IR photothermoelectric (PTE) polarization-sensitive photodetectors for both linear and circular polarization detections. The design principle not only provides a powerful platform to transfer the polarization-sensitive optical response to an electrical signal readout, but also can be readily applied to other wavelength regions, such as the visible, the near-infrared, and the terahertz, because the response of our device is not limited by the bandgap of active semiconductors. The chiral plasmonic metamaterials designed for SoP-dependent absorption create localized temperature gradients through photothermal effect under uniform illumination, which in turn generates a polarization-resolved photovoltage response in 2D thermoelectric materials. In addition, the polarity transition for both linear and circular polarization detections can be realized by spatial and geometric configurations of chiral metamaterials within the device channel. Furthermore, as a proof-of-concept demonstration of superiority of the balanced photodetectors ($PR = -1$, $g = \infty$), a properly designed device with three-ports for full-Stokes detection is demonstrated, and a polarization imaging demonstration is presented with the developed device. Our results show a filterless, uncooled, bandgap-independent, devisable wavelength-specific, configurable, and polarization-dependent photodetection mechanism based on the combination of nanophotonic structures and thermoelectric materials on an integrated chip. It offers a promising platform for optoelectronic applications and opens possibilities for the next-generation mid-infrared photodetection, polarimetry, and imaging technologies.

## Results and discussion

### Resonant PTE response of designed detectors

The architecture of our proposed mid-IR PTE detector (Fig. 1a) consists of a two-dimensional thermoelectric material, two electrodes, and a chiral plasmonic metamaterial comprising a periodic array of chiral plasmonic meta-atoms. The chiral plasmonic molecule is made from a Z-shaped gold (Au) nanostructure on top of a dielectric spacer ($Al_2O_3$) and a thick Au backplate. Figure 1b shows the unit cells of left-handed (LH) and right-handed (RH) metamaterials for preferred LCP and RCP light absorber. For a typical metamaterial under normal incident radiation with a transverse magnetic polarization, the simulated absorption spectrum shows a wavelength peak at around $5.3\,\mu m$ as shown in Fig. 1c (blue). Cross-sections of electric field distribution and power absorption distribution with on- and off-resonance modes are shown in Fig. 1d, e. The on- and off-resonance modes correspond to the maximum and the minimum absorptions at wavelengths of 5.3 and $5.1\,\mu m$, respectively. As compared with the off-resonance mode, the on-resonance mode induces larger electric fields in the Au antenna, giving rise to a higher absorption of light. On the other hand, the main absorption of light occurs at the Au antennas. When the metamaterials is illuminated by a mid-IR light beam with a diameter of $100\,\mu m$ and a power of 5 mW, the photothermal effect[34] increases the temperature of the Au antenna. Figure 1c (red) plots the increased temperature $\Delta T$ as a function of the wavelength. The maximum $\Delta T$ located at resonance wavelength can reach up to 3 K. More importantly, the absorption spectrum shows a similar response (i.e., the blue and red curves in Fig. 1c nearly overlap), indicating a linear relation between the absorption and $\Delta T$. Such a linear relation is crucial for transferring polarization-dependent optical absorption to polarization-resolved electrical signal outputs.

As an example of the PTE response[35,36], we consider a device with left half-channel covered by metamaterials. The channel consists of a 2D thermoelectric material with a Seebeck coefficient ($S$) of $-200\,\mu V\,K^{-1}$ and a thermal conduction of $4.5\,W\,m^{-1}\,K^{-1}$. A temperature source designed according to the results of photothermal simulations is used as the input. The temperature distribution in our device is simulated with a consideration of the heat conductance, radiation, and convection. An example illustrating the temperature distribution in our device is shown in Fig. 1f. Similarly, when the device with a channel length of $30\,\mu m$ is illuminated by a mid-IR light beam with a diameter of $100\,\mu m$ and a power of 5 mW, the left half-channel covered by metamaterials exhibits a higher temperature than the right half-channel without metamaterials. As a result, a temperature gradient is built up within the device channel as shown in Fig. 1g (red). Such a temperature gradient can generate a potential difference between the source (S) and the drain (D) electrodes, giving rise to a photovoltage response ($V_{ph}$) of $170\,\mu V$ shown in Fig. 1g (blue), which can be calculated by $V_{ph} = -S\Delta T$.

### Photothermal response of chiral plasmonic metamaterials

To investigate the photothermal response of the chiral plasmonic metamaterials, full-wave electromagnetic simulations were performed on the Z-shaped Au antenna array. First, structural parameters as indicated in Fig. 1b are obtained using global optimization. The optical absorption of chiral metamaterials can be tuned across a broadband mid-IR regime ($4-8\,\mu m$) as shown in Supplementary Fig. 1 and Supplementary Table 1. Owing to the sandwich construction of metamaterials, a Fabry–Pérot-like cavity is formed between the Z-shaped antennas and the ground plane, leading to a multiple reflection under a wide-field illumination[37]. On the other hand, because the planar metamaterials are anisotropic and lossy, linear polarization conversion is introduced and results in destructive or constructive interference for the incident light with different SoP[38]. The simulated absorptions for different metamaterials shows not only a linear polarization dependency, but also a circular polarization dependency with a

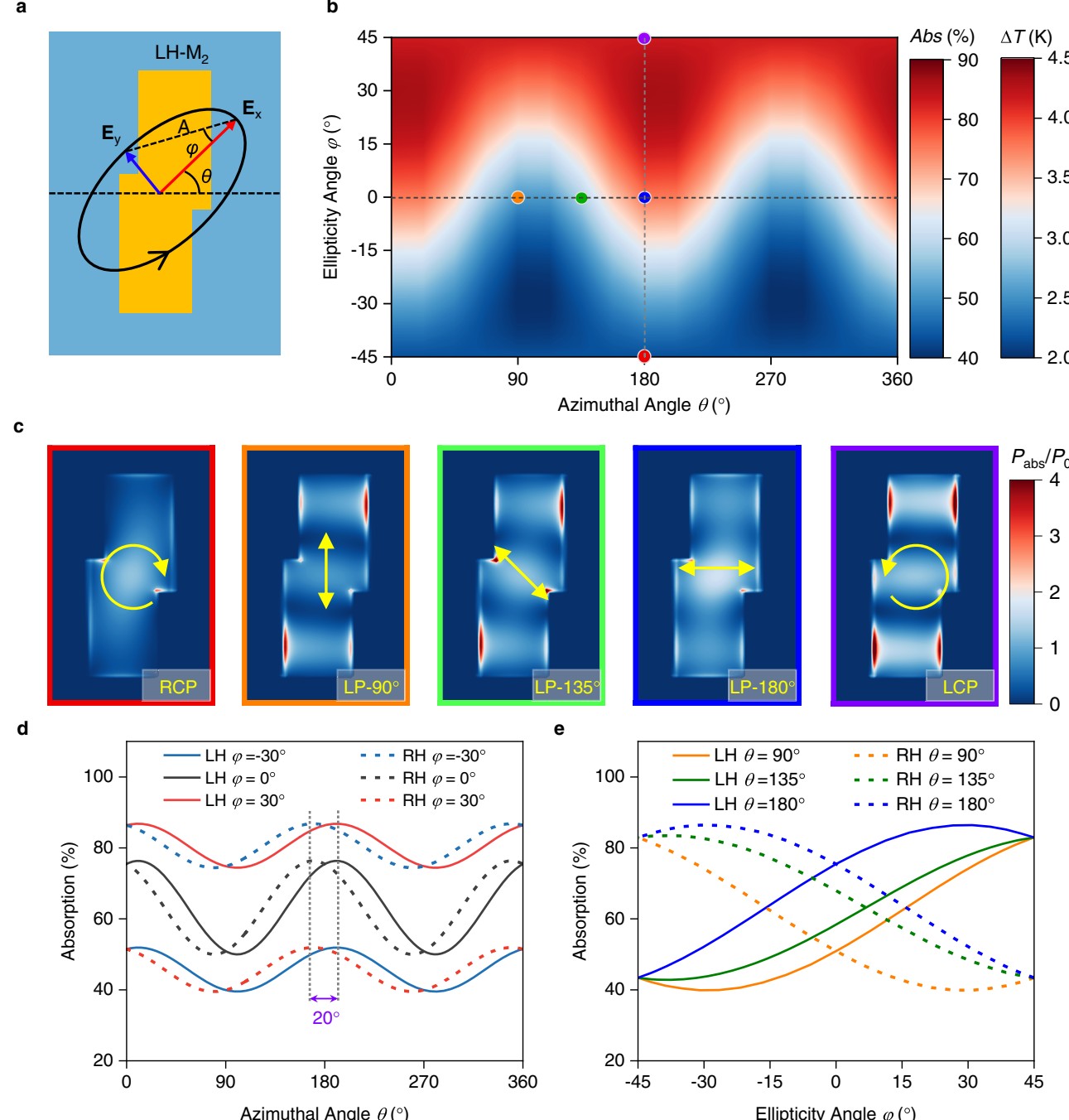

**Fig. 2 | Polarization-dependent photothermal effect of the chiral metamaterials. a** Schematic of the LH-$M_2$ metamaterial and incident polarized light described by the geometrical parameters of the ellipse. $E_x$ and $E_y$ denote the electric field vector along the semi-major axis and semi-minor axis, respectively. $\theta$ denotes the angle between the semi-major axis of the polarization ellipse and the x-axis. $\varphi$ denotes the ellipticity angle and equals to arctan $E_x/E_y$. **b** 2D contour map of both simulated absorbance and induced temperature increase for different light polarization status. **c** Corresponding power absorption density normalized by $P_0$ for different polarization status marked by red, orange, green, blue, and purple dots in (**b**), respectively. **d, e** The simulated absorbance as a function of azimuthal angle at different ellipticity angles (**d**) and as a function of ellipticity angle $\varphi$ at different azimuthal angles (**e**) for both LH and RH metamaterials. Two dashed lines in (**d**) show a phase shift of 20° between two absorption peak value positions of LH and RH metamaterials.

circular dichroism (CD) of 50% in a broadband mid-IR regime (Supplementary Fig. 2). Furthermore, we experimentally verified the simulated polarization-dependent optical absorption of the as-designed metamaterials. Here, three typical metamaterials ($M_1$, $M_2$, and $M_4$) are fabricated and their polarization-dependent absorptions are measured (Supplementary Figs. 3–5). The experimentally measured optical absorption spectra of both the LH and RH metamaterials show the same polarization dependence as the simulation results. For

the LH metamaterial, the absorption at the resonance wavelength is a cosine function of the linear polarization angle, which fits well with simulation results. In addition, the absorptions are significantly different under the LCP and RCP light illuminations, resulting in a large CD of about 30%. Such a value is high enough to distinguish the LCP and RCP light in practical devices.

To further investigate the SoP-dependent photothermal response of metamaterials, we study a typical LH-$M_2$ metamaterial with a

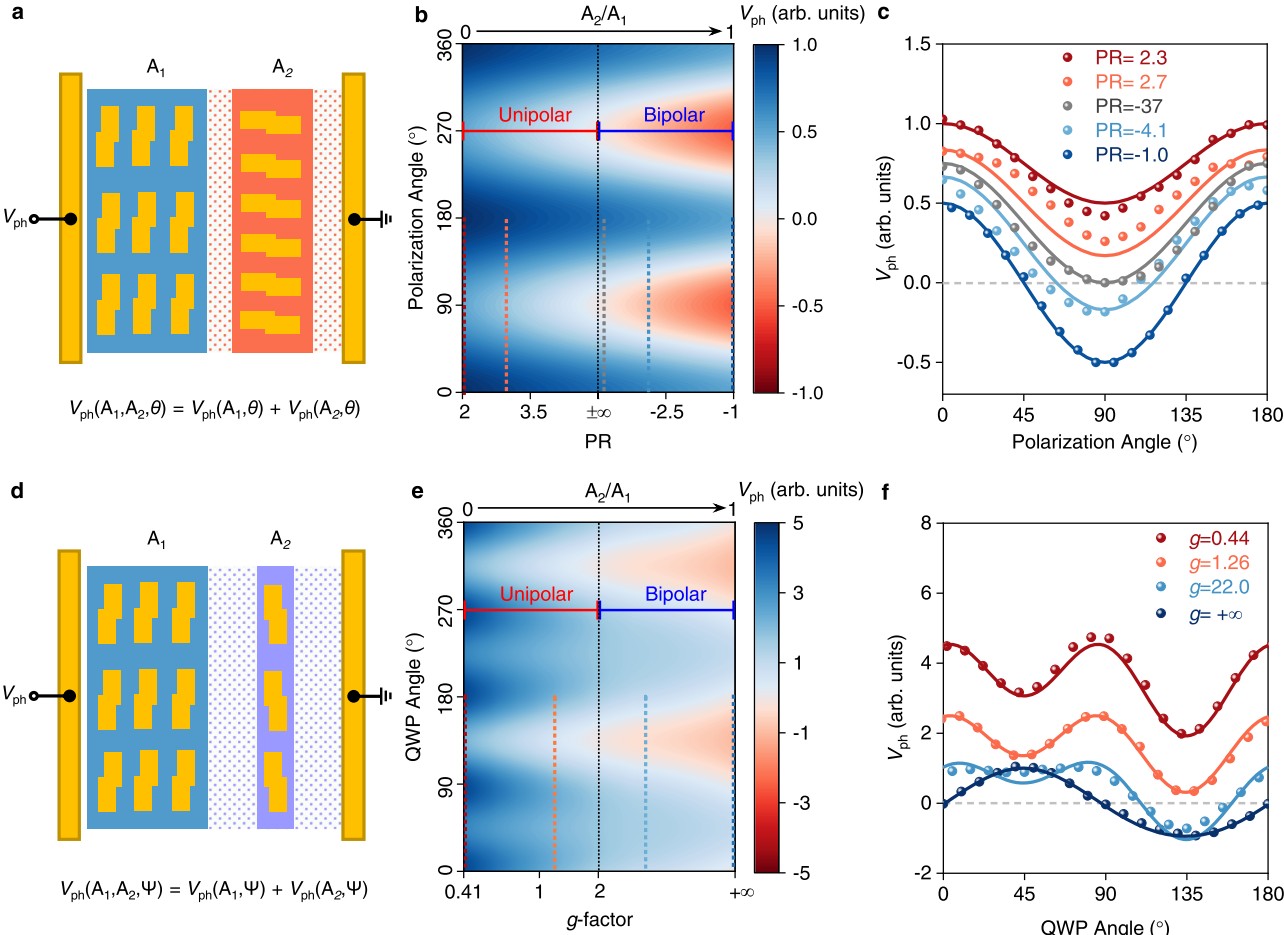

**Fig. 3 | Configurable polarity transition of linear and circular polarization dependence. a** Calculation of the linear polarization angle $\theta$-dependent photoresponse, $V_{ph}$ (A$_1$, A$_2$, $\theta$) with different distribution area ratio (A$_2$/A$_1$) of two nanoantennas with an orientation angles of 90°. **b** Calculation of the linear polarization angle $\theta$-dependent photoresponse $V_{ph}$ (A$_1$, A$_2$, $\theta$) with respect to the ratio of A$_2$/A$_1$ exhibits a polarization ratio (PR) transition from unipolar regime to bipolar regime. The color bar shows the normalized photoresponse. **c** Simulated (lines) and measured (symbols) photoresponses of five typical devices with their polarization ratio (PR) values indicated by the dashed lines in (**b**) with the same colors. **d** Calculation of the quarter-wave plate (QWP) angle $\psi$-dependent photoresponse, $V_{ph}$ (A$_1$, A$_2$, $\psi$) with different distribution area ratio (A$_2$/A$_1$) of two nanoantennas (LH and RH). **e** Calculation of the QWP angle $\psi$-dependent photoresponse, $V_{ph}$ (A$_1$, A$_2$, $\psi$) with respect to the ratio of A$_2$/A$_1$ exhibits a $g$-factor transition from unipolar regime to bipolar regime. The color bar shows the normalized photoresponse. **f** Simulated (lines) and measured (symbols) photoresponses of four typical devices with their $g$-factor values indicated by the dashed lines in (**e**) with the same colors.

resonance peak at 5.3 μm as another example. As shown in Fig. 2a, the SoP of incident light is described in a general way by the geometrical parameters of an ellipse. The $\theta$ and $\varphi$ denote the azimuthal and ellipticity angle, respectively, and the sign of $\varphi$ indicates the chirality of polarization (positive→left-handed, negative→right-handed). Figure 2b shows the contour map of the simulated absorption as a function of azimuthal angle $\theta$ and ellipticity angle $\varphi$. Because of the linear relation between the absorption (*Abs*) and the temperature increase ($\Delta T$), $\Delta T$ shows a similar dependence on the azimuthal angle $\theta$ and ellipticity angle $\varphi$. The absorption difference between the maximum and the minimum absorptions can reach about 50%, leading to a temperature increase difference of about 2.5 K. For a fixed ellipticity angle $\varphi$, both the absorption and the temperature increase $\Delta T$ can fit well with a cosine function of the azimuthal angle $\theta$ with a weighted shift factor given by ellipticity angle $\varphi$. In detail, the absorption *Abs* can be calculated by the following fitted formula:

$$Abs = a + b\cos\left(2(\theta+10)\right) \tag{1}$$

$$\begin{pmatrix} a \\ b \end{pmatrix} = \begin{pmatrix} a_1 \\ b_1 \end{pmatrix} + \begin{pmatrix} a_2 \\ b_2 \end{pmatrix}\left(\sin(2\varphi)\cos(2\varphi)\right) \tag{2}$$

where, $a$ indicates the azimuthal angle independent constant background absorption, and $b$ indicates the amplitude of $\theta$-resolved absorption component with a fixed $\varphi$. The $a_1$ and $a_2$ represent the constant component and the amplitude of the $\varphi$-resolved $a$ component as a function of sine, respectively, and $b_1$ and $b_2$ represent the constant component and the amplitude of the $\varphi$-resolved $b$ component as a function of cosine, respectively. The 10° indicates the relative angle between the equivalent orientation and the long axis direction of the Z-shaped nanoantenna.

$$\begin{pmatrix} a_1 \\ b_1 \end{pmatrix} \sim \begin{pmatrix} 63.15 \\ 0 \end{pmatrix} \tag{3}$$

$$\begin{pmatrix} a_2 \\ b_2 \end{pmatrix} \sim \begin{pmatrix} 19.70 \\ 13.16 \end{pmatrix} \tag{4}$$

here, the values in the matrix are extracted by fitting the simulation results (the details of the extraction process are presented in Supplementary Note 1). In particular, full-wave electromagnetic simulations have been performed for five polarization states indicated by colored dots in Fig. 2b. As shown in Fig. 2c, the high polarization-dependent

absorptions come from the destructive or constructive interferences of the incident light beams through the linear polarization conversion (Supplementary Fig. 6). The simulated temperature increase, $\Delta T$, also shows a high polarization dependency and a linear relation to the incident light intensity (Supplementary Fig. 7).

We then consider the SoP-dependent optical responses for both LH and RH metamaterials. As shown in Fig. 2d, for a fixed ellipticity angle $\varphi$, absorptions for both LH and RH metamaterials follow a similar cosine-dependence on the azimuthal angle with only a 20° phase shift. For the elliptically polarized light with different fixed azimuthal angle $\theta$, the absorption shows a reverse trend for LH and RH metamaterials when the ellipticity angle $\varphi$ changes from −45° to 45° (Fig. 2e). It is noticed that the change of absorption with the ellipticity angle $\varphi$ is non-monotonic and is also dependent on the azimuthal angle $\theta$. These optical absorption properties of the designed metamaterials directly correspond to the temperature increase $\Delta T$ of Au antenna, thereby directly determining the photovoltage response in the metamaterial-mediated devices.

## Configurable polarity transition of polarization dependence

The high SoP-dependent optical absorption directly leads to a polarization-resolved PTE response in the detector integrated with chiral plasmonic metamaterials[9,39]. To experimentally demonstrate this, we first fabricated devices using different 2D thermoelectric materials including graphene (Gr)[40,41], black phosphrous (BP)[13], and palladium selenide (PdSe₂)[42,43] nanoflakes as the active thermoelectric materials according to the configuration shown in Fig. 1f. All the fabricated devices show a linear polarization angle-dependent photovoltage ($V_{ph}$) response, and hence, can also be applied to distinguish the LCP and RCP lights (see Supplementary Fig. 8), indicating a high tolerance for selection of active materials for our proposed polarization-sensitive photodetection mechanism. We note that the polarization-sensitive photoresponse comes from the plasmonic chiral metamaterials, but not from the intrinsic anisotropy of the active thermoelectric materials owing to the uniform illumination. Particulary, the working wavelength for BP photodetection can be extended to 5.3 μm, which is beyond its conventional cut-off wavelength of about 4.1 μm according to its bandgap at 0.3 eV[12]. This indicates that the operation wavelength of our proposed approach is no longer limited by the bandgap of the active material. On the other hand, the photoresponse of the PdSe₂-based device is higher than that of Gr- and BP-based devices because of its higher Seebeck coefficient. In our following experimental demonstration, we use PdSe₂ nanoflakes as the active materials and M₂ metamaterials unless noted otherwise. In addition, the working wavelength of the proposed detectors can be designed by using appropriate metamaterials, revealing a devisable wavelength photodetection mechanism (Supplementary Fig. 9). We further investigated the device performance at room temperature using a PdSe₂-based device integrated with LH-M₂ metamaterials. As shown in Supplementary Figs. 10–12, the wavelength-dependent photovoltage responsivity is in accordance with the absorption spectrum of the metamaterials, which further verifies the resonant PTE response of our device. Moreover, the detector exhibits a high responsivity up to 3.6 V W⁻¹, a short response time of 76 μs corresponding to a −3 dB bandwidth of 1.1 kHz, a low dark noise spectrum down to 35 nV Hz⁻¹/² corresponding to a noise-equivalent power of 9.7 nW Hz⁻¹/², a specific detectivity of $2.5 \times 10^5$ Jones, and good repeatability and stability at room-temperatures. Furthermore, the device exhibits a lower photoresponse at lower temperatures (Supplementary Fig. 12), which results from the low-temperature gradient owing to the efficient heat dissipation or high thermal conductivity at low temperatures[44,45].

Leveraging on the linear and circular polarization dependence of the metamaterials mediated photoresponse, we first design devices with a geometrically configurable polarity transition for linear polarization-sensitive detection, by using LH metamaterials distributed in the left-half (A₁) and right-half (A₂) channel with a fixed relative orientation angle $\alpha = 90°$ (Fig. 3a). The polarization-dependent photovoltage responses $V_{ph}$ for different distribution area ratio (A₂/A₁) are calculated and plotted in Fig. 3b showing that the PR changes from 2 to −1 with a polarity transition (from unipolar to bipolar) when A₂/A₁ changes from 0 to 1. This is a clear evidence that the linear polarization-sensitive detector exhibits a geometrically configurable polarity transition. Our five fabricated devices, with A₂/A₁ = 0, 0.33, 0.5, 0.67, and 1, experimentally verify the configurability of PR (see Supplementary Figs. 13, 14). Normalized experimental polarization-resolved photoresponses, with PR = 2.3, 2.7, −37, −4.1, and −1, show good agreements with the calculated results in the configurable polarity transition (Fig. 3c). We also investigate the configurable polarization dependence of designed devices with A₂/A₁ = 1 by changing the relative orientation angle $\alpha$ from 0° to 90° (Supplementary Fig. 14). As shown in Supplementary Fig. 15, the calculated polarization-dependent photovoltage responses exhibit an $\alpha$-independent PR, but with a phase shift of the maximum response angle. Our fabricated devices with $\alpha = 0°$, 45°, and 90°, show a good agreement in phase shift with the calculated results indicating a configurable polarization dependence, which is useful for full-Stokes detection in practical applications.

Thanks to the configuration flexibility of metamaterials, the proposed resonant PTE response also enables us to realize circular polarization detection. We then investigated the geometrically configurable polarity transition for the circular polarization-sensitive detection, which is crucial for realizing direct detection of chirality and ellipticity, simultaneously. Here, a series of devices with LH and RH metamaterials distributed in the left-half (A₁) and right-half (A₂) channels with various distribution area ratios A₂/A₁ were designed (Fig. 3d). With a fixed A₁, the g-factor changes from about 0.41 to +∞ as the ratio A₂/A₁ changes from 0 to 1, with the polarity transition occurring at $g = 2$. When the g-factor is in the range of 0–2, the device shows a unipolar circular polarization-dependent photoresponse. While when the g-factor is in the range of $g > 2$, the device shows a bipolar photoresponse (Fig. 3e). In the same way, a series of devices with various distribution area ratios A₂/A₁ were fabricated and their circular polarization-dependent photoresponses were measured by rotating a quarter-wave plate (QWP) (see Supplementary Figs. 16, 17). We also compare our experimental results of the QWP angle-dependent photoresponses with the calculation in Fig. 3f. Four fabricated devices with A₂/A₁ = 0, 0.33, 0.67, and 1 exhibit a g-factor of 0.44, 1.26, 22, and +∞, respectively. This indicates a configurable polarity transition for the circular polarization-sensitive detection. It is worth noting that, owing to the phase shift of linear polarization-dependent absorption between LH and RH metamaterials as shown in Fig. 2d, the relative orientation angle $\alpha$ should be set as 20° to eliminate the linear polarization component from the QWP angle-dependent photoresponse. This results in a pure circular polarization-resolved photovoltage response, which can be fitted by a standard Sine function (Supplementary Fig. 17). We note that the change of QWP angle from 45° to 135° corresponds to an ellipticity angle change from 45° (LCP) to −45° (RCP) along with chirality and ellipticity changes. The photovoltage response of the device with A₂/A₁ = 1 and $\alpha = 20°$ shows a monotonic relation with QWP angle in the range of 45°–135°, and a sign-flipping at 90°, indicating the ability of simultaneous detections of the chirality and the ellipticity. In addition, the QWP angle-dependent photovoltage response shows a robustness to the linear polarization angle $\theta$ (Supplementary Fig. 18). Therefore, by geometrical configuration such as changing the distribution area ratio and the relative orientation angle, the designed device can not only distinguish the LCP and RCP light, but also directly detect the chirality and ellipticity, simultaneously. More detailed comparisons with the existing linear and circular

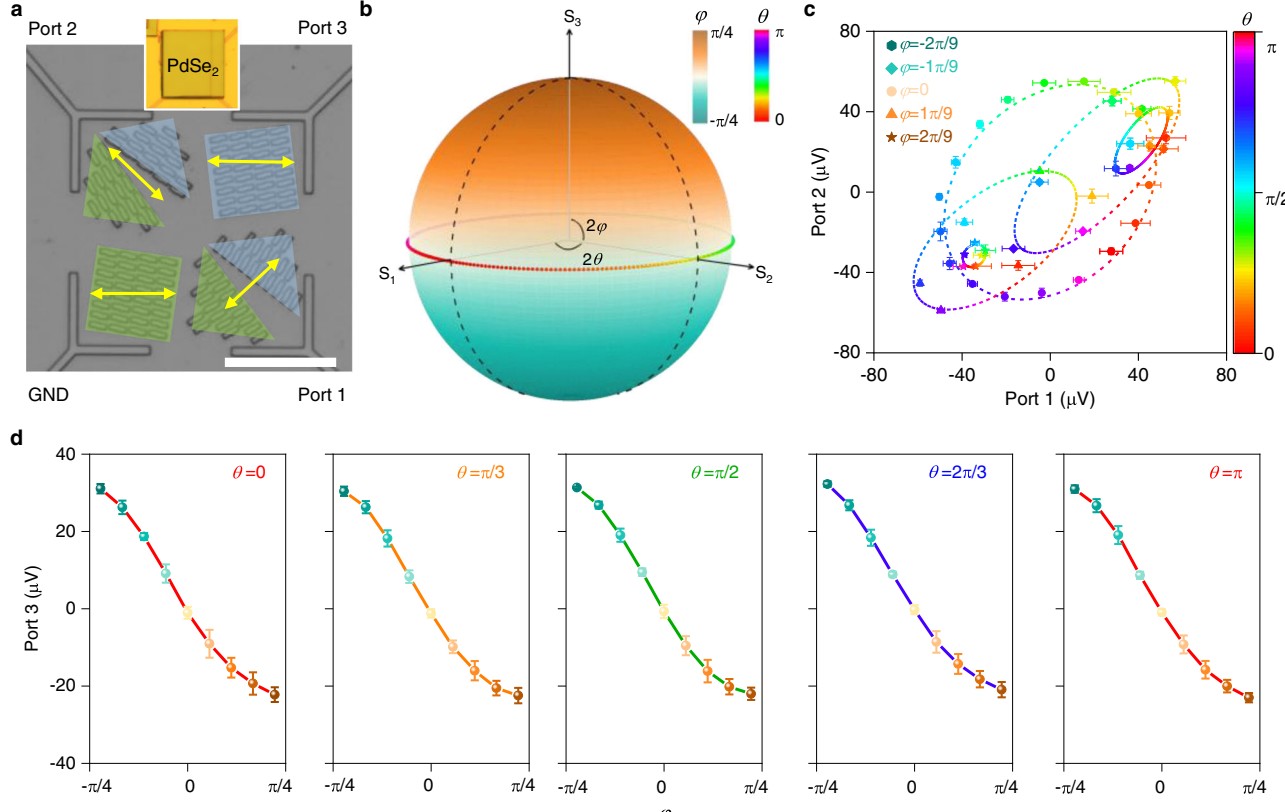

**Fig. 4 | Full-Stokes polarization detection. a** Optical image of metamaterials and electrodes of the three-ports device for full-Stokes polarization detection. The GND port is used as the ground terminal for other three port outputs. The yellow arrows indicate the equivalent orientation of metamaterials in each part, and the green and blue colors indicate LH and RH chiral metamaterials, respectively. Scale bar: 25 μm. Inset is the optical image of the device. **b** Schematic illustration of the full-Stokes information in a Poincaré sphere. $S_1$ and $S_2$ are the linear parameters that characterize the direction of the linear component. $S_3$ is the circularly polarized parameter that quantifies the circular component. **c** Two-dimensional plot of Port 1 and Port 2 under different azimuthal angle $\theta$ and ellipticity angle $\varphi$. Symbols are measured data, which are presented as mean values ± SD, $n = 4$ replicated measurements. Dashed lines are fitting cures. **d** The Port 3 outputs as a monotonic function of ellipticity angle $\varphi$ under different azimuthal angle $\theta$. Symbols are measured data, which are presented as mean values ± SD, $n = 4$ replicated measurements.

polarization-sensitive photodetectors are provided in Supplementary Tables 2, 3.

## Full-Stokes detection in a three-ports device

To demonstrate the usefulness of the bipolar devices for both linear and circular polarized light detection, we designed a three-ports photodetector with the advantages of resolving the SoP of different incident polarized light (Fig. 4a). For an arbitrary SoP, it can be described by using the geometrical parameters of an ellipse including the amplitude A, the azimuthal angle $\theta$, and the ellipticity angle $\varphi$. On the other hand, the SoP can also be described by a Stokes vector in a Poincare sphere as shown in Fig. 4b. Here, four Stokes parameters include the total intensity $S_0$, two linear components $S_1$ and $S_2$, and the chiral component $S_3$. For a fully polarized light, the relation between Stokes parameters and geometrical ellipse parameters can be described with following formula:

$$S_0 = A^2 \tag{5}$$

$$S_1 = A^2 \cos 2\theta \cos 2\varphi \tag{6}$$

$$S_2 = A^2 \sin 2\theta \cos 2\varphi \tag{7}$$

$$S_3 = A^2 \sin 2\varphi \tag{8}$$

Generally, six individual pixels are required to fully retrieve the full-Stokes parameters. Here, taking advantages of the configurability of our proposed resonant PTE detection mechanism, a three-ports device has been designed to extract the geometrical ellipse parameters instead of the three Stokes parameters ($S_1$, $S_2$, and $S_3$). It's worth noting that, owing to the polarization-sensitive photoresponses of the three-ports outputs, the Stokes parameter $S_0$ would not be extracted using the designed three-ports device. On the other hand, the SoP in this work is focused on fully polarized light but not on partially polarized or unpolarized light. As shown in Fig. 4a, the GND port is used as the ground terminal for other three ports. The relative orientation angle of LH and RH metamaterials distributed in triangle shape is set to be 20° to align their equivalent orientations as discussed above. With a consideration of the existence of a mirror symmetry in linear polarization-dependent photoresponse in the range of 0–180°, at least two outputs with a phase shift not equal to 90° are required to unambiguously detect the polarization angle[46,47]. Therefore, the equivalent orientation (yellow arrows) for each part is well designed and the LH (green) and RH (blue) metamaterials distributions are also well designed. The Ports 1 and 2 in our device are used for extracting the azimuthal angle $\theta$, and Port 3 is used for extracting the ellipticity angle $\varphi$. To verify the unambiguous detection of SoP in our three-ports device, the photovoltage outputs of three ports are measured at different polarization angles obtained by changing the HWP and QWP angles but constant incident power (Supplementary Fig. 19). Based on the theoretical analyses and experimental results, the azimuthal angle $\theta$ and ellipticity

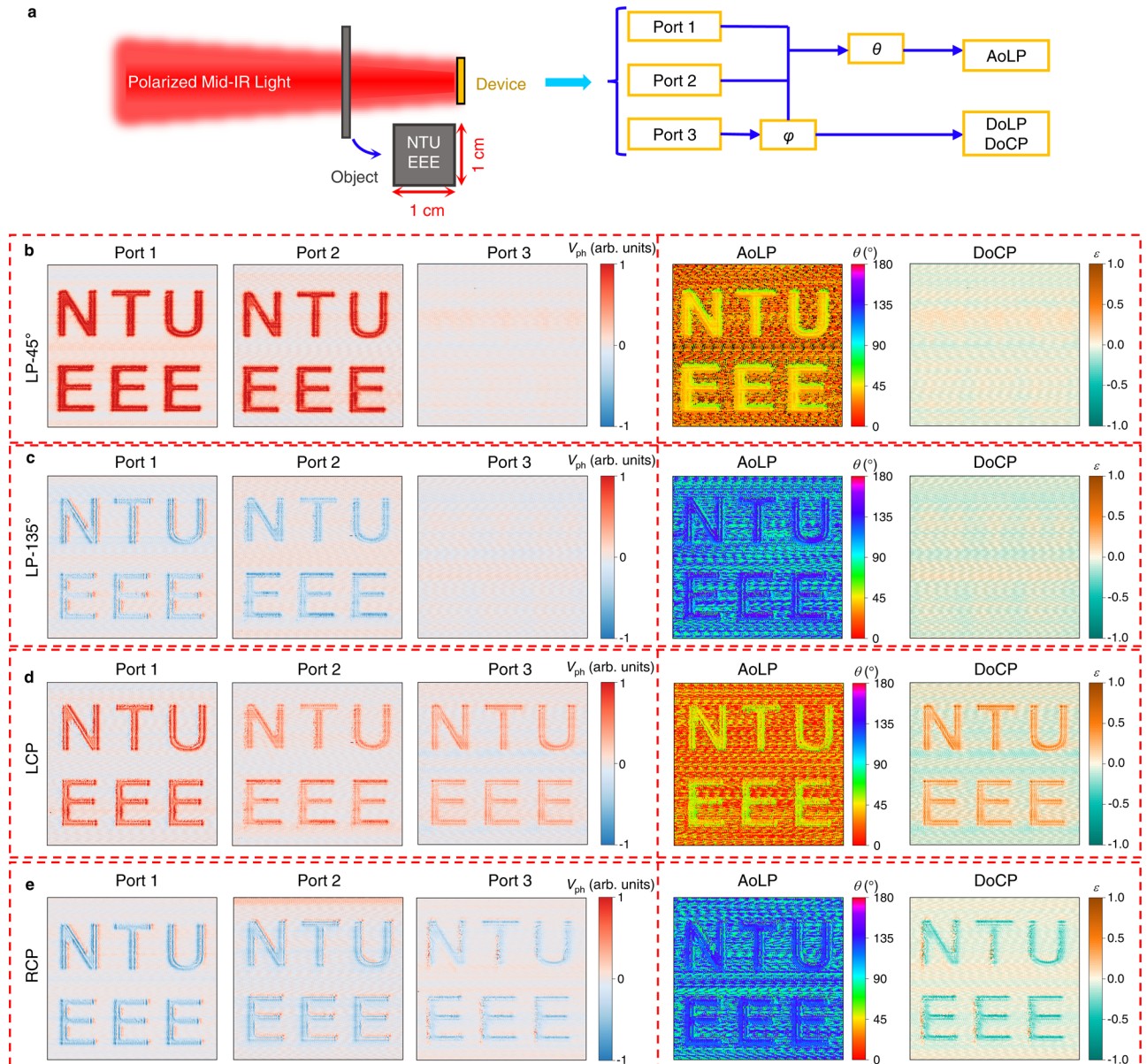

**Fig. 5 | Polarimetric imaging applications of the three-ports device.**
**a**, Schematic of the polarimetric imaging measurement system and the mechanism for the calculation of angle of linear polarization (AoLP), degree of linear polarization (DoLP) or degree of circular polarization (DoCP). The inset is the schematic of the object with text pattern NTU and EEE. **b**–**e** Measured Imaging results of three ports (Port 1, Port 2, and Port 3), and calculated imaging results of AoLP and DoCP under linear polarized ((**b**) LP-45° and (**c**) LP-135°) and circular polarized ((**d**) LCP and (**e**) RCP) light illuminations.

angle $\varphi$ can be calculated as:

$$\theta = \frac{1}{2}\left(\tan^{-1}\frac{P_1 + P_3}{P_2 + P_3} + 45\right) \quad \text{when } P_2 < 0 \qquad (9)$$

$$\theta = \frac{1}{2}\left(\tan^{-1}\frac{P_1 + P_3}{P_2 + P_3} + 45\right) + 90 \quad \text{when } P_2 > 0 \qquad (10)$$

$$\varphi = \tan^{-1}\frac{P_3}{C} \qquad (11)$$

where $C$ refers to the maximum photovoltage output under LCP or RCP light illumination with a typical incident power (the derivation of these expressions is presented in the Supplementary Note 2). Figure 4c shows the 2D plots of the measured photovoltages of the Port 1 and the Port 2 with the dots colored and shaped based on the azimuthal angle $\theta$ and the ellipticity angle $\varphi$ of the incident polarized light. For a typical

ellipticity angle $\varphi$, the (Port 1, Port 2) pairs move anticlockwise along a closed elliptical curve. In addition, when the ellipticity angle $\varphi$ changes from −45° to 45°, the elliptic center moves from the first quadrant to the third quadrant through the origin point. Although the (Port 1, Port 2) pairs show a dependence on both the azimuthal angle $\theta$ and the ellipticity angle $\varphi$, another photovoltage output is necessary to unambiguously detect the SoP. This is because there are some intersections between the elliptic curves for different ellipticity angles $\varphi$. Figure 4d depicts the photovoltage response as a function of ellipticity angle $\varphi$ under different azimuthal angles. The monotonic relationship between the Port 3 output and ellipticity angle $\varphi$ ranging from −45° to 45° enables us to directly readout the ellipticity angle $\varphi$ even for different azimuthal angles $\theta$. Furthermore, based on the relation between the incident light power and the photovoltage outputs of the three ports, we can obtain the amplitude A with calibration (Supplementary Fig. 20). To evaluate the polarimetry accuracy of our three-ports device, we calculate the deviation of the three Stokes parameters

based on a set of measurements. The result is shown in Supplementary Fig. 21. The average measurement errors of $S_1$, $S_2$, and $S_3$ are 14.2%, 15.2%, and 5.4%, respectively. The relative higher measurement errors for $S_1$ and $S_2$ than that for $S_3$ can be attributed to the nonimmune photoresponses of Port 1 and Port 2 against the circular polarization. On the other hand, the imperfect fabrication of metamaterials, the imperfect Gaussian distribution of laser beam, the inaccuracy of input light polarization, and so on, will also introduce measurement errors for the Stokes parameters.

## Polarization imaging applications

To highlight the practical usage of our polarimeter with the compact and simplified configuration, a polarization imaging application is demonstrated using the three-ports device. The polarimetric imaging enables us to obtain important information about the surfaces of targets by detecting the spatially and temporally varying SoP of light. As shown in Fig. 5a, the polarized mid-IR light is illuminated onto the photodetector through an object with patterned NTU and EEE letters. Based on the photovoltage signal outputs from the three ports, both the ellipticity angle $\varphi$ and the azimuthal angles $\theta$ can be calculated. In addition, the angle of linear polarization (AoLP), degree of linear polarization (DoLP), or degree of circular polarization (DoCP) can be calculated by data processing. Particularly, we consider the fully polarized incident light in this work, and the DoCP and DoLP can be calculated as[21,48]:

$$AoLP = \theta \qquad (12)$$

$$DoCP = \frac{2\varepsilon \sin(2\varphi)}{1 + \varepsilon^2} \qquad (13)$$

$$DoLP = \sqrt{1 - DoCP^2} \qquad (14)$$

where $\varepsilon$ refers to the ellipticity and is calculated by: $\varepsilon = \tan \varphi$. Figure 5b–e show the imaging results from three ports under a 5.3 μm light illumination with LP-45°, LP-135°, LCP, and RCP polarization status, respectively. For different polarizations of the incident light, all three ports (left three columns in Fig. 5b–e) can obtain clearer polarization imaging results and exhibit a typical combining form, which is a one-to-one correspondence to the SoP. In addition, both the corresponding AoLP and DoCP imaging results can also be obtained with data processing, as shown in the right two columns in Fig. 5b–e. Compared with conventional division-of-focal-plane polarimeters that require at least four signal outputs, our proposed three-ports device including two bipolar linear polarization detectors and one bipolar circular polarization detector shows potential in high-resolution and fast polarization imaging applications owing to its compact configuration and simplified signal processing procedure. Therefore, the proposed polarimeter shows great potentials in mid-IR polarimetric imaging. For practical imaging applications, the size of single pixel, the crosstalk between adjacent pixels, and the scalability of the read-out circuit, also need to be considered.

In summary, we have presented mid-IR polarization-sensitive PTE detectors with several advantages such as filterless, uncooled, bandgap-independent, tailorable in the operating wavelength, compact, and configurable in polarization-dependence. Combining the SoP-dependent optical response and the design flexibility of the chiral plasmonic metamaterials with the two-dimensional thermoelectric materials, polarity transition can be realized for both the linear and the circular polarizations. Leveraging on the bipolar photoresponse, a three-ports device is demonstrated to unambiguously detect the SoP of incident light with a compact device configuration and a simplified signal process procedure. Last, we demonstrate the infrared polarimetric imaging capability such as AoLP and DoCP imaging of the proposed detection strategy by using a three-ports polarimeter. This shows great potential for the emerging optical technologies in the mid-IR range, such as polarimetric imaging, molecule sensing, fiber optics and/or free-space communications.

## Methods

### Simulation

The simulation of optical responses and photothermal effect of the chiral plasmonic metamaterials were done using Lumerical FDTD Solutions and HEAT packages. In all simulations, a single unit structure and a plane wave light source were used. Periodic boundary conditions and perfectly matched layers were used in the x&y boundaries and z-boundaries, respectively. The simulated structure consists of a silicon substrate, $SiO_2$ (285 nm thickness), gold backplate (200 nm thickness), $Al_2O_3$ dielectric space layer (200–270 nm), gold antennas (50 nm thickness) and air. The power density absorption was calculated using the equation: $P_{abs} = 1/2\omega\varepsilon''|\mathbf{E}|^2$, where $\omega$ is the light frequency and $\varepsilon''$ is the imaginary part of the dielectric function. For the photothermal effect simulation in Heat package, an import heat source according to the optical absorption data obtained from FDTD simulation result is used as the heat input. One temperature monitor placed surrounding the antenna is used to record the temperature profile. Considering the weak thermal flow through convection and diffusion at the boundaries, different boundary conditions were applied for different surfaces/interfaces in our simulations, i.e., (1) a convection of 10 W/(m²·K) is applied at the top surfaces of Au nanostructures and the $Al_2O_3$; (2) a heat flux of 10 W/m² is set across the interfaces between Au and $Al_2O_3$, Au and $SiO_2$, $SiO_2$ and Si to simulate the weak thermal flow in the solid interfaces; (3) the temperature of the bottom boundary of the simulation domain is fixed as room temperature (293 K). For the simulation of temperature distribution of device with large scale (2 × 2 mm²), COMSOL Multiphysics software with Heat Transfer Modules was used. The fixed temperature thermal boundary condition is applied at the surface of antennas according to the result from the photothermal effect simulation in the HEAT package, while other boundaries, which are far away from the Au nanostructures, has a temperature fixed as the room temperature (293 K).

### Device fabrication

As the first step to device fabrication, a 200-nm-thick gold thin film and a $Al_2O_3$ dielectric space layer with a typical thickness (200–270 nm) were first deposited onto a heavily p-doped silicon wafer grown with 285 nm thermal $SiO_2$ using e-beam evaporation. Then, electrodes and gold nanoantenna arrays were patterned on the chips using standard electron-beam lithography followed by thermal deposition of 5-nm-thick Cr and 50-nm-thick Au and lift-off process (submerging samples in acetone for 1 h). Thereafter, two-dimensional thermoelectric materials, such as graphene (Gr), black phosphorus (BP), and PdSe₂ nanoflakes were mechanically exfoliated from their bulk crystal and then were transferred onto the special position of the chip with electrodes and metamaterials by a dry-transfer method.

### Characterization

The optical absorption and CD spectrum were obtained using a Fourier Transform Infrared spectrometer (FTIR, Bruker) with a microscope (Thermo Fisher). Linear and circular polarized lights are generated by using a linear polarizer and a quarter-wave plate. For the reflection spectra, the same sample without chiral metamaterials was used as the reference. The transmission is negligible due to the optically thick gold backplate. The absorbance spectra of the metamaterials were calculated by using equation: $Abs = (1 - Ref) \times 100\%$. The polarized photoresponse is measured by using a homemade photocurrent measurement system where the infrared light with different polarization status is obtained from a serious quantum cascade lasers (Daylight Solutions, MIRcat) with high linear polarization purity (>100:1) and

tunable wavelength in the range of 4–8 μm combining a serious half-wave plate and quarter-wave plate, and then focused on the samples using a zinc selenide IR focusing lens with a focal length of 50 mm. The generated photovoltage was then recorded by a highly sensitive source meter unit (Keysight, B2912A). For the low-temperature photoresponse measurement, the device is mounted in a vacuum cryostat with a temperature controller. Here, we have selected three typical wavelengths in this work based on the operating wavelength of the half-wave plate and quarter-wave plate, 4.5 μm (Thorlabs, WPLH05M-4500 and WPLQ05M-4500), 5.3 μm (Thorlabs, WPLH05M-5300 and WPLQ05M-5300), and 7.0 μm (Edmund, #85-121 and #85-114). The voltage noise is measured by using a lock-in amplifier (Zurich Instruments, HF2LI). The voltage data were collected within 1 min with a time constant of 1 s and a typical internal reference frequency. The low-frequency (<1 kHz) temporal photoresponse for response speed analysis was measured using an oscilloscope (Keysight, DSOX3054T) with the signal pre-amplified (Stanford Research Systems, SR570) and an optical chopper (Thorlabs, MC1F10A).

## Polarization imaging measurements

The imaging measurements are carried out by using a homemade imaging system. The polarized infrared light with different polarization status is obtained as mentioned above. An optical mask with "NTU EEE" letter was put into the light path and its position is controlled by two step motors along the x-axis and y-axis. By changing the mask location, the photoresponse signals from the three-ports device are amplified using a preamplifier and recorded by an oscilloscope.

## Data availability

Relevant data supporting the key findings of this study are available within the article and the Supplementary Information file. All raw data generated during the current study are available from the corresponding author Q.J.W. upon request. Source data are provided with this paper.

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

## Acknowledgements

This research was supported by National Research Foundation Singapore programme (NRF-CRP18-2017-02 (Q.J.W.) and NRF-CRP22-2019-0007 (Q.J.W.)) and A*STAR grant number A18A7b0058 (Q.J.W.), A20E5c0095 (Y.L., Q.J.W.), and A2090b0144 (Q.J.W.).

## Author contributions

M.D., Y.L., and Q.J.W. conceived the project. M.D. did the theoretical analysis and numerical simulation with assistance from B.Q., M.Y., and S.H. M.D. did sample fabrication with assistance from C.W. M.D. carried out the device characterization with assistance from F.W. All authors discussed the results. M.D., Y.L., and Q.J.W. wrote the manuscript with comments from all authors. Q.J.W. supervised the project.

## Competing interests

The authors declare no competing interests.
