## [Peer Review File · Nature Communications]

On-chip mid-infrared photothermoelectric detectors for full-Stokes detectionREVIEWER COMMENTS

Reviewer #1 (Remarks to the Author):

The manuscript reports the fabrication of On-chip mid-infrared photothermoelectric detectors. The results seem interesting and are worthy of reference and creativity. I recommend the publication on Nat. Commun. However, several comments have to address here :

1. The temperature is not always constant at such a small scale, there must have thermal fluctuations. So can we consider the Neumann boundary condition? Can the left-handed and right-handed metamaterials still maintain a relatively big difference when there is weak thermal flow at the boundary?
2. The incident light power used in this experiment is large. Can the device be applied to weak-light detection, such as nW-level light, and what is the specific detectivity of the device?
3. The bandgap of PdSe₂ is directly related to its thickness. It is better to use an atomic force microscope to characterize the thickness of PdSe₂. I noticed that you had adopted the Fabry-Pérot resonant cavity to enhance the absorption of the device, so how does the thickness of PdSe₂ and Al₂O₃ material affect the whole cavity? In other words, what thickness of PdSe₂ and Al₂O₃ is best for the device response?
4. In the illustration part of Figure 2a, the font in the description of ϕ is inconsistent with the font of the text, please modify it.
5. Below room temperature, the heat loss of the device should be severe. Can you characterize the performance of the device at different temperatures? Can the device work at different temperatures?
6. For the imaging part, if there is a strong left-handed light mixed with weak right-handed light, will the detector be able to distinguish a clear pattern? If not, is there a way to solve the problem?

Reviewer #2 (Remarks to the Author):

The authors report on the development of an on-chip polarization-sensitive photodetector. There are two key elements in this polarization detector. The first one is a chiral metasurface that has polarization-dependent heating. Polarization-dependent heating of chiral metasurfaces in mid-IR has been demonstrated a while ago (no references presented by the authors), see, for example, Nature Comm. 7:12045 (2016). The second element -- very well known and widely used -- is a thermoelectric 2D material exhibiting strong Seebeck effect. It is not entirely clear to me that combining these two previously demonstrated effects to make a detector is sufficient for a publication in Nature Communications. If so: the authors should make a clear case for it.

Further on the innovation side, I was specifically asked to compare the concept behind this work with three other recent publications: (a) Nat. Photon. 15, 614 (2021), (b) ACS Photonics 5, 4283 (2018), and (c) ACS Nano 14, 16634 (2020). Ref.(a) was referenced by the authors, although not entirely appropriately, in my opinion. References (b) and (c) were not referenced. I will start with Ref.(a), which was cited as Ref.31 in the manuscript. It was stated in the manuscript that the dis-symmetry coefficient ranging from minus to plus infinity was not demonstrated in Ref.(31). While technically correct, their structures certainly allow distinguishing between RCP and LCP light because they are inherently chiral. So I am not sure if there is a conceptual difference between the two. The authors would have to explain the difference.

Ref.(b) also measured all Stokes parameters in mid-IR by integrating a plasmonic metasurface with a 2D material. While the work used a slightly different approach (instead of using four distinct metasurfaces, it was using a single graphene-tunable metasurface), but the outcome is similar. Similarly, Ref.(c) used four distinct chiral metasurfaces integrated with graphene. Although Ref.(c) did not rely on polarization-

dependent heating because it was designed to operate at telecom wavelengths, the outcome is still the same: full Stokes polarimetry. Therefore, all three references should be cited, and the novelty of this new work explained in the context of (a-c).

Reviewer #3 (Remarks to the Author):

The authors demonstrated an on-chip polarization detection device for mid-IR wavelengths based on plasmonic chiral metamaterials and 2D thermoelectric materials. It is an interesting device concept for addressing the challenges in polarization detection in mid-IR wavelength ranges. I recommend major revisions before publication.

1. What material did the authors use for the device they presented? It was quite vague in the manuscript. The authors only mentioned ... "two-dimensional thermoelectric materials, such as graphene (Gr), black phosphorus (BP), and PdSe₂ nanoflakes were mechanically exfoliated from their bulk crystal and then were transferred onto the special position of the chip..." Figure 1 shows 2D thermal electric material while Figure 4a shows PdSe₂. If the authors tried all three and would like to emphasize their method is applicable to all, please provide a clear description of the corresponding results (in the supplementary information), and comment on their differences so that the readers understand how to choose proper materials for different wavelengths and/or applications.
2. The author claimed that compared with the conventional spatial multiplexing method which requires at least four individual pixels to fully retrieve the full-Stokes parameters, their only require a three-port device for polarization detection because of their unique device design. However, the conventional method is used to obtain all four polarimetric parameters including S₀, the intensity; while the proposed method could not measure intensity but instead requires constant intensity with a three-ports device. Moreover, the authors should note that their method would only measure purely polarized light, not including partially polarized and unpolarized light.
3. Equation 2 on page 8 has some values which are obtained by fitting simulation results of some specific designs. The authors should generalize their analysis using variables with clearly defined meanings in the equation. Then explain how to extract these coefficients in the manuscript or the method section.
4. How did the authors obtain equations 7, 8 9 on page 12? I could not find the corresponding discussion and derivation. The authors could include the explanation in the method section if it is too lengthy.
5. Figure 4 shows the results regarding the stokes parameter detection. But the plots do now show clearly the measurement accuracy for stokes parameters, for comparison with other techniques presented in the literature. It would be much more straightforward to show the measurement results and extracted errors for stokes parameters, S₁, S₂, and S₃, on these plots. The authors should also add some discussions about the measurement errors and indicate fundamental limitations, e.g., cross-talk, of their method and the corresponding impact on the measurement error if there are any.
6. What is the operating temperature of the device? Is it at room temperature? Would the device work better at a lower temperature? What is the detection speed (time for measuring one polarization state)?
7. The authors used an object on a motorized 2D stage to complete a polarization mapping. This is not the same as what people usually refer to as mid-IR imaging. From a single element polarization detection device to a real polarimetric imaging sensor, one needs to consider the pixel size, the cross-talk between adjacent pixels, and the scalability of the read-out circuit.

Reply to the Reviewers' Comments

We are grateful to the kind and constructive advice on our manuscript which has been submitted to Nature Communications (ID: NCOMMS-22-14122. Title: On-chip mid-infrared photothermoelectric detectors for full-Stokes detection). In response to these comments, we have carried out further experiments, provided necessary discussions and analyses, and revised the manuscript and the supplementary information accordingly. We believe that we have addressed all the comments raised by the reviewers.

The corresponding amendments addressing the referees' concerns have been marked with yellow background for easy reference in the revised manuscript (Revised Manuscript with marks) and revised Supplementary Information (Revised Supplementary Information with marks). Our replies to the comments by the editor and reviewers and the corresponding changes are listed below.

The following are details of our point-by-point responses to reviewers' comments:

Reviewer #1:

Comments:

The manuscript reports the fabrication of On-chip mid-infrared photothermoelectric detectors. The results seem interesting and are worthy of reference and creativity. I recommend the publication on Nat. Commun. However, several comments have to address here:

Response to reviewer: We really thank the reviewer for commenting that our results are interesting and are worthy of reference and creativity.

Comment 1: The temperature is not always constant at such a small scale, there must have thermal fluctuations. So can we consider the Neumann boundary condition? Can the left-handed and right-handed metamaterials still maintain a relatively big difference when there is weak thermal flow at the boundary?

Response to reviewer: We thank the reviewer for pointing out this aspect which helps improve the clarity of our manuscript. In fact, we considered the Neumann boundary conditions in our simulations and the photothermal effect of the metamaterials was

simulated using the Heat package in Lumerical. The periodic boundary conditions were applied along the x&y directions. On the other hand, the two surfaces perpendicular to the z direction are treated with different boundary conditions, i.e. the bottom surface of the simulation domain has a temperature fixed as the room temperature (293 K), while a convection of $10 \text{ W}/(\text{m}^2 \cdot \text{K})$ was set at the top surfaces of Au nanostructures and the Al_2O_3 . Note that, to simulate the weak thermal flow in the solid interfaces, the interface between Au and Al_2O_3 , Au and SiO_2 , SiO_2 and Si were set to be the heat flux boundary condition with a heat flux of $10 \text{ W}/\text{m}^2$. The temperature distribution of the device is simulated using COMSOL Multiphysics software in a large scale ($2 \times 2 \text{ mm}^2$). In other words, it is reasonable to fix the temperature of the boundary as the room temperature because the boundary condition is robust against the photothermal effect of Au nanostructures (Nat Nanotech 2017, 12(8): 770-775). As indicated by simulation results shown in Supplementary Information Figure S7, it still maintains a relatively big difference for LCP and RCP light illuminations when there is weak thermal flow at the boundaries. To clearly describe the simulation parameters, we have added an additional description in the Method section, also copied below:

On page 16 of the revised manuscript.

Simulation.One temperature monitor placed surrounding the antenna is used to record the temperature profile. Considering the weak thermal flow through convection and diffusion at the boundaries, different boundary conditions were applied for different surfaces/interfaces in our simulations. i.e. 1. a convection of $10 \text{ W}/(\text{m}^2 \cdot \text{K})$ is applied at the top surfaces of Au nanostructures and the Al_2O_3 ; 2. a heat flux of $10 \text{ W}/\text{m}^2$ is set across the interfaces between Au and Al_2O_3 , Au and SiO_2 , SiO_2 and Si to simulate the weak thermal flow in the solid interfaces 3. The temperature of the bottom boundary of the simulation domain is fixed as room temperature (293 K). For the simulation of temperature distribution of device with large scale ($2 \times 2 \text{ mm}^2$), COMSOL Multiphysics software with Heat Transfer Modules was used. The fixed temperature thermal boundary condition is applied at the surface of antennas according to the result from the photothermal effect simulation in the HEAT package, while other boundaries, which are far away from the Au nanostructures, has a temperature fixed as the room temperature (293K).

Comment 2: The incident light power used in this experiment is large. Can the device be applied to weak-light detection, such as nW-level light, and what is the specific detectivity of the device?

Response to reviewer: We appreciate the referee for this comment. In fact, the mW-level

light used in our work indicates the total laser power which is globally illuminated on the whole sample, and the power on the detector is in the μW -level range as shown in Figure S20 of the Supplementary Information. According to the noise measurement and responsivity of a half-mediated device based on PdSe_2 nanoflakes, a noise-equivalent power (NEP) of $9.7 \text{ nW Hz}^{-1/2}$ is achieved at a high modulation frequency over 1.1 kHz, corresponding to a nW-level light detection ability of our device. The specific detectivity of our device is calculated to be about 2.5×10^5 Jones at room temperature by using $D^* = A^{1/2}/\text{NEP}$, where A is the device area ($30 \times 20 \mu\text{m}^2$). To clearly describe the performance of our device, we have added the specific detectivity in the main text, also copied below:

Lines 1-4, on page 10 of revised manuscript.

..... Moreover, the detector exhibits a high responsivity up to 3.6 V W^{-1} , a short response time of $76 \mu\text{s}$ corresponding to a -3dB bandwidth of 1.1 kHz, a low dark noise spectrum down to $35 \text{ nV Hz}^{-1/2}$ corresponding to a noise-equivalent power of $9.7 \text{ nW Hz}^{-1/2}$ and a specific detectivity of 2.5×10^5 Jones, and good response repeatability and stability at room-temperature.....

Comment 3: The bandgap of PdSe_2 is directly related to its thickness. It is better to use an atomic force microscope to characterize the thickness of PdSe_2 . I noticed that you had adopted the Fabry–Pérot resonant cavity to enhance the absorption of the device, so how does the thickness of PdSe_2 and Al_2O_3 material affect the whole cavity? In other words, what thickness of PdSe_2 and Al_2O_3 is best for the device response?

Response to reviewer: We thank the reviewer for the constructive and useful suggestion. The thickness-dependent bandgap of PdSe_2 has been widely studied and demonstrated in theoretical calculations and experimental measurements (J. Am. Chem. Soc., 2017, 139, 14090–14097; npj 2D Mater. Appl., 2022, 6, 1; ACS Nano, 2020, 14, 4963–4972). According to the literature, the bandgap decreases from 1.3 eV to 0 eV when the thickness changes from monolayer to bulk materials, and the bandgap changes slightly when the layer number is over 24 (corresponding to $\sim 14 \text{ nm}$). However, in our case, the device response is directly related to the photothermal effect of metamaterials and Seebeck effect of thermoelectric materials. In other words, the impact of thickness on the Seebeck coefficient of PdSe_2 can no longer be ignored. According to previous studies (e.g. Adv. Func. Mater. 2020, 30, 2004896), the thinner PdSe_2 nanoflakes possess higher Seebeck coefficient owing to the quantum confinement effects, corresponding to a higher photothermoelectric response. In addition, this trend has also been observed in our previous work (ACS Nano 2022, 16, 1, 295–305). In this work, the thickness of PdSe_2 nanoflakes used for device fabrication are in the range of 100-200 nm, which has little impact on the

device responses.

As for the resonant cavity, the optical absorption of metamaterials is influenced by the thickness of Al_2O_3 . As shown in Figure S1 in the revised Supplementary Information, the thickness of Al_2O_3 influences not only the resonance peak, but also the absorption intensity for different polarization states. To achieve a high optical absorption contrast for both linear and circular polarized light, the thicknesses of Al_2O_3 is set to be 200 nm for the metamaterials with a peak resonance at 5.3 μm . Furthermore, the thicknesses of Al_2O_3 are optimized for different resonance peaks as listed in Table S1 in the Supplementary Information. On the other hand, thickness of PdSe_2 nanoflake has negligible impact on the resonant cavity because it is set on the top of metamaterials, which can be further verified by the experimental results as shown in Supplementary Information Figure S10. The photoresponse of the device exhibits the same spectral profile as the optical absorption of the pristine metamaterials.

In conclusion, the thickness of Al_2O_3 is optimized by FDTD solutions by considering a robust polarization-dependent absorption, while the thickness of PdSe_2 nanoflakes have negligible impact on the device responses when it is in the range of 100-200 nm in this work.

On page 5 of the revised Supplementary Information.

Fig. S1 Effect of dielectric spacer (Al_2O_3) thickness on the optical absorption of metamaterials. a-d, The absorption spectrum of LH- M_2 metamaterials with different dielectric spacer thicknesses for different polarization states.

Comment 4: In the illustration part of Figure 2a, the font in the description of φ is inconsistent with the font of the text, please modify it.

Response to reviewer: We thank the reviewer for pointing out this issue. The corresponding font of φ has been modified in the revised manuscript.

Comment 5: Below room temperature, the heat loss of the device should be severe. Can you characterize the performance of the device at different temperatures? Can the device work at different temperatures?

Response to reviewer: We thank the reviewer for the constructive and useful suggestion. We measured the photoresponse of the device at different temperatures, and added

corresponding description in the revised manuscript, also copied below:

Lines 5-8, on page 10 of the revised manuscript.

.....Furthermore, the device exhibits a lower photoresponse at lower temperatures (Supplementary Figure S12), which results from the low temperature gradient owing to the efficient heat dissipation or high thermal conductivity at low temperatures.^{44, 45}

On page 16 of revised Supplementary Information.

Fig. S12 Temperature dependent photoresponse. a, The photoresponses at different temperatures. b, The temperature-dependent photovoltages with an incident light power of $38 \mu W$.

Comment 6: For the imaging part, if there is a strong left-handed light mixed with weak right-handed light, will the detector be able to distinguish a clear pattern? If not, is there a way to solve the problem?

Response to reviewer: We appreciate the referee for this interesting comment. In fact, there are two cases of mixture between a strong left-handed light and a weak right-handed light when we consider the propagating direction. One case is that the strong left-handed light and weak right-handed light are propagating along the same direction. And the other case is that the strong left-handed light and weak right-handed light are propagating along different directions.

For the former case, in principle, a strong left-handed light mixed with a weak right-handed light is equivalent to an elliptically polarized light with a fixed left-handed chirality and a fixed azimuthal angle. If we describe this light by using the geometrical parameters of an ellipse, the fixed azimuthal angle θ can be extracted by combining the Port 1, Port 2, and Port 3 outputs, and the ellipticity angle φ can be extracted directly from Port 3 output.

For the latter case, a strong left-handed light mixed with a weak right-handed light can be equivalent to an elliptically polarized light with a fixed left-handed chirality and a space-variable azimuthal angle within the cross-section. If we describe this light by using the geometrical parameters of an ellipse, the space-variable azimuthal angle θ can be extracted by combining Port 1, Port 2, and Port 3 outputs, and the fixed ellipticity angle φ can be extracted directly from Port 3 output because the Port 3 output in our device is immune to the azimuthal angle θ .

Therefore, as for the scenario suggested by the reviewer, the detector is still able to distinguish a clear pattern for the imaging part.

Reviewer #2:

Comments:

The authors report on the development of an on-chip polarization-sensitive photodetector. There are two key elements in this polarization detector. The first one is a chiral metasurface that has polarization-dependent heating. Polarization-dependent heating of chiral metasurfaces in mid-IR has been demonstrated a while ago (no references presented by the authors), see, for example, Nature Comm. 7:12045 (2016). The second element -- very well known and widely used -- is a thermoelectric 2D material exhibiting strong Seebeck effect. It is not entirely clear to me that combining these two previously demonstrated effects to make a detector is sufficient for a publication in Nature Communications. If so: the authors should make a clear case for it.

Response to reviewer: We are grateful to the reviewer for the review and appreciate the reviewer's constructive suggestions/comments that have greatly helped us to improve the quality and depth of our work. From the perspective of polarization-sensitive photodetectors, the novelty of our work comes from two major aspects.

1. From the aspect of realization of on-chip polarization-sensitive photodetectors, our work provides an alternative way to transfer the optical polarization-sensitive response to an electrical signal readout, thereby achieving an ultracompact on-chip polarimeter. This strategy is different from previous methods in following points:

1-1) Bandgap-robust spectral response. Previous methods to realize polarization-sensitive photodetectors are typically based on the optical anisotropy or chirality from the natural semiconductors or the artificial plasmonic metamaterials integrated semiconductors. The spectral response of these detectors is bandgap-limited because of their photoconductive or photovoltaic response mechanisms. In contrast, the spectral response of the photothermoelectric response mechanism proposed in our work is bandgap-robust.

1-2) Devisable wavelength-specific response. Although, the resonance wavelength of the artificial plasmonic metamaterials is devisable, it must be matched with the bandgap of semiconductor for the photoconductive/photovoltaic response mechanism. In contrast, the spectral response of the photothermoelectric response mechanism proposed in our work is devisable in a broadband spectrum without the limitation of cut-off wavelength caused by the bandgap.

2. From the aspect of performance of the on-chip polarization-sensitive photodetectors, our work realizes both the linear and circular polarization detection with configurable polarity, as well as state of polarization detection in a three-ports device. This achievement is well

justified by following two points:

2-1) Both linear and circular optical responses of the chiral metamaterials. To realize both linear and circular polarization dependent photoresponse, the chiral metamaterials should not only possess chiral optical absorption, but also anisotropic optical absorption. Therefore, the Z-shaped chiral metamaterials are designed and used in this work, as compared to other spiral or gammadion-shaped chiral metamaterials demonstrated in the literature which only possess chiral optical responses.

2-2) The temperature-distribution dependent photoresponse of thermoelectric materials. To realize configurable polarity photoresponse, the photocurrent or photovoltage should be reconfigurable by the state of polarization. In this work, the polarization-resolved photothermal effect induced temperature-distribution is geometrically configurable, thereby leading to a reversible photovoltage response through the Seebeck effect of the thermoelectric materials.

In the revised version of the manuscript, we added additional explanation to better show the novelty of our work, also copied below:

Lines 7-10, on page 5 of the revised manuscript.

.....The design principle not only provides a powerful platform to transfer the polarization-sensitive optical response to an electrical signal readout, but also can be readily applied to other wavelength regions, such as the visible, near-infrared, and terahertz, because the response of our device is not limited by the band gap of active semiconductors.....

Lines 18-21, on page 5 of the revised manuscript.

..... Our results show, for the first time, a filterless, uncooled, bandgap-robust, devisable wavelength-specific, and configurable polarization-dependent photodetection mechanism based on the combination of nanophotonic structures and thermoelectric materials on an integrated chip.....

Comment 1: Further on the innovation side, I was specifically asked to compare the concept behind this work with three other recent publications: (a) Nat. Photon. 15, 614 (2021), (b) ACS Photonics 5, 4283 (2018), and (c) ACS Nano 14, 16634 (2020). Ref. (a) was referenced by the authors, although not entirely appropriately, in my opinion. References (b) and (c) were not referenced. I will start with Ref. (a), which was cited as Ref.31 in the manuscript. It was stated in the manuscript that the dis-symmetry coefficient ranging from minus to plus infinity was not demonstrated in Ref. (31). While technically correct, their structures certainly allow distinguishing between RCP and LCP light because

they are inherently chiral. So I am not sure if there is a conceptual difference between the two. The authors would have to explain the difference.

Ref.(b) also measured all Stokes parameters in mid-IR by integrating a plasmonic metasurface with a 2D material. While the work used a slightly different approach (instead of using four distinct metasurfaces, it was using a single graphene-tunable metasurface), but the outcome is similar. Similarly, Ref.(c) used four distinct chiral metasurfaces integrated with graphene. Although Ref.(c) did not rely on polarization-dependent heating because it was designed to operate at telecom wavelengths, the outcome is still the same: full Stokes polarimetry. Therefore, all three references should be cited, and the novelty of this new work explained in the context of (a-c).

Response to reviewer: We are thankful to the reviewer for the kind recommendation and constructive comments. According to the reviewer's suggestions, all three references have been cited, and the novelty of our work has been explained in the context of these references.

Comparing with Ref.(a), our work has the following two innovations:

First, the photoresponse mechanism of our work is based on the Seebeck effect of thermoelectric materials driven by the temperature gradient induced by photothermal effect of metamaterials. Based on this working mechanism, the selection of active thermoelectric materials is no longer limited by their bandgap which was demonstrated by our experimental results. However, the photoresponse mechanism of Ref.(a) is based on the local excited photocurrents driven by Seebeck coefficient gradient at the metal-graphene interface, then establishing an electric field throughout the gapless material that drives the ambient carriers to contact electrodes. Therefore, this kind of work mechanism in Ref.(a) requires a semimetal material without bandgap, which brings limitation in active material choices.

Second, we realized geometrically configurable polarity transition for the circular polarization-sensitive detection. With a well-designed device configuration, we can realize simultaneous detections of the chirality and ellipticity with immunity against the linearly polarized light. As for Ref.(a), their simulation results show that their detectors cannot realize the simultaneous detection of the chirality and ellipticity because of the non-monotonic relationship between the photovoltage and phase difference, although they can distinguish the LCP and RCP by the sign of photoresponse (as shown in their Supplementary Figure 26).

Comparing with Ref.(b), our work has the following innovation:

Ref.(b) uses a chiral metamaterial integrated graphene modulator with a separate polarizer and a photodetector in free space to realize full-Stokes detection. This approach belongs to free space detection in a bulk optics approach and requires a high spatial degree of freedom, and hence it does not belong the integrated on-chip detection categories. On the other hand, the Stokes parameters of the incident light are extracted by sweeping the gate voltage applied to the device and subsequent fitting of the measured reflected intensities. Such a method needs to take more steps and normally more measurement time when compared with our ‘snap-shot’ on-chip detection mechanism. In contrast, our work demonstrates a fully on-chip polarimeter and only three electrical readouts are needed to extract the state of polarization, which offers a route to achieve compact polarimetry for polarized light detection in spatial and temporal variation, as well as polarization imaging with ultra-high pixel density.

Comparing with Ref.(c), our work has the following innovation:

Ref.(c) uses four chiral metamaterial integrated Si photodetectors to realize full-Stokes detection. The working mechanism is based on the plasmonic enhanced photoconductive effect of Si, which transfers polarization-sensitive absorption enhancement of metamaterials/semiconductor to photocurrent readout through the plasmonic resonance between metamaterials and semiconductors. Such an approach has major limitation on the working wavelength as the working wavelength is not only related to the resonance peak of metamaterials, but also limited by the bandgap of active semiconductors. However, as mentioned above, our work mechanism is more robust and makes the selection of active thermoelectric materials more flexible which is not limited by their bandgap.

Based on above comparisons and discussions, we added additional explanation in the revised version of the manuscript to better explain the novelty of our work, also copied below:

Lines 9-11, on page 3 of the revised manuscript.

Full SoP detection is highly desirable for many applications. Such bulky optical systems by using free-space polarizer have intrinsic drawbacks such as limited speed, limited accuracy, and incomplete polarization state detection, making them unsuitable for these applications⁹

Lines 5-10, on page 4 of the revised manuscript.

However, most of previous detectors relied on photoconductive or photovoltaic effect, which requires a match between the resonant wavelength of plasmonic metamaterials and the bandgap of semiconductors.^{7, 28} Therefore, an efficient way to transfer the strong anisotropy

and chirality to electrical readouts without the working wavelength limitation by the bandgap of active materials is highly desired.

Lines 28-32, on page 4 of the revised manuscript.

To increase the PR value, the bipolar linear-polarization-sensitive photodetectors have been recently realized by introducing the Demer effect modulated by the photonic mechanism and hot-carrier mechanism by integrating nanoantenna on graphene.^{14, 33} As a result, the PR can be controlled to have values in the range of $(1 \rightarrow \infty / -\infty \rightarrow -1)$ with a transition from positive to negative. However, such a realization only demonstrated the bipolar linear polarization detection, while the polarity transition for circular polarization detection has not been realized till today. On the other hand, the robust detection of CPL with immunity against the ubiquitous unpolarized and linearly polarized light has not been realized.

Refs. a-c have also been cited as follows:

7. Li L, Wang J, Kang L, Liu W, Yu L, Zheng B, *et al.* Monolithic Full-Stokes Near-Infrared Polarimetry with Chiral Plasmonic Metasurface Integrated Graphene–Silicon Photodetector. *ACS Nano* 2020, **14**(12): 16634-16642.

9. Jung M, Dutta-Gupta S, Dabidian N, Brener I, Shcherbakov M, Shvets G. Polarimetry Using Graphene-Integrated Anisotropic Metasurfaces. *ACS Photonics* 2018, **5**(11): 4283-4288.

33. Wei J, Xu C, Dong B, Qiu C-W, Lee C. Mid-infrared semimetal polarization detectors with configurable polarity transition. *Nat Photonics* 2021, **15**(8): 614-621.

Reviewer #3:

Comments:

The authors demonstrated an on-chip polarization detection device for mid-IR wavelengths based on plasmonic chiral metamaterials and 2D thermoelectric materials. It is an interesting device concept for addressing the challenges in polarization detection in mid-IR wavelength ranges. I recommend major revisions before publication.

Response to reviewer: We are thankful to the reviewer for the kind recommendation and suggestions. We also appreciate reviewer for commenting that our device concept is interesting for addressing the challenges in polarization detection in mid-IR wavelength ranges.

Comment 1: What material did the authors use for the device they presented? It was quite vague in the manuscript. The authors only mentioned ... “two-dimensional thermoelectric materials, such as graphene (Gr), black phosphorus (BP), and PdSe₂ nanoflakes were mechanically exfoliated from their bulk crystal and then were transferred onto the special position of the chip...” Figure 1 shows 2D thermal electric material while Figure 4a shows PdSe₂. If the authors tried all three and would like to emphasize their method is applicable to all, please provide a clear description of the corresponding results (in the supplementary information), and comment on their differences so that the readers understand how to choose proper materials for different wavelengths and/or applications.

Response to reviewer: We thank the reviewer for pointing out the aspects which helps us improve our manuscript. In fact, we tried all three different 2D materials with Seebeck effect to emphasize our method is applicable to all. According to reviewer’s suggestion, we added a clear corresponding description in the revised manuscript, also copied below:

Lines 15-28, on page 9 of the revised manuscript.

.....All the fabricated devices show a linear polarization angle dependent photovoltage (V_{ph}) response and hence, can also be applied to distinguish the LCP and RCP lights (Supplementary Figure S7), indicating a high tolerance for active materials of our proposed polarization-sensitive photodetection mechanism. We note that the polarization-sensitive photoresponse comes from the plasmonic chiral metamaterials, but not from the intrinsic anisotropy of the active thermoelectric materials owing to the uniform illumination. Particularly, the working wavelength for BP photodetection can be extended to 5.3 μm , which is beyond its conventional cut-off wavelength 4.1 μm according to its bandgap of 0.3 eV,¹² indicating that

our proposed approach is no longer limited by the bandgap of the active material. On the other hand, the photoresponse of PdSe₂ based device is higher than that of Gr and BP based devices because of its higher Seebeck coefficient. In our following experimental demonstration, we use PdSe₂ nanoflakes as the active materials and M₂ metamaterials unless noted otherwise.....

Comment 2: The author claimed that compared with the conventional spatial multiplexing method which requires at least four individual pixels to fully retrieve the full-Stokes parameters, their only require a three-port device for polarization detection because of their unique device design. However, the conventional method is used to obtain all four polarimetric parameters including S₀, the intensity; while the proposed method could not measure intensity but instead requires constant intensity with a three-ports device. Moreover, the authors should note that their method would only measure purely polarized light, not including partially polarized and unpolarized light.

Response to reviewer: We thank the reviewer for the question. As all photovoltage outputs for three ports in our device are all polarization-sensitive, the light intensity information indeed is not extracted from the measurement. We agree with the reviewer that our method is focused on the measurement of purely polarized light. To clearly describe the limitation of our method, we added corresponding description in the revised manuscript, also copied below:

Lines 10-17, on page 12 of the revised manuscript.

.....Generally, six individual pixels are required to fully retrieve the full-Stokes parameters. Here, taking advantages of the configurability of our proposed resonant PTE detection mechanism, a three-ports device has been designed to extract the geometrical ellipse parameters instead of three Stokes parameters (S₁, S₂, and S₃). It's worth noting that, owing to the polarization-sensitive photoresponses of the three-ports outputs, the Stokes parameter S₀ can't be extracted using the designed three-ports device. On the other hand, the SoP in this work is focused on fully polarized light but not include partially polarized and unpolarized light.....

Comment 3: Equation 2 on page 8 has some values which are obtained by fitting simulation results of some specific designs. The authors should generalize their analysis using variables with clearly defined meanings in the equation. Then explain how to extract these coefficients in the manuscript or the method section.

Response to reviewer: We thank the reviewer for the constructive suggestion. According to the reviewer's suggestion, we generalized the variables with clearly defined meanings and explained how to extract the coefficients in the revised Supplementary Information. The corresponding description has been added in the revised manuscript, also copied below:

Lines 7-21, on page 8 of the revised manuscript.

.....For a fixed ellipticity angle φ , both the absorption and temperature increase ΔT can fit well with a cosine function of the azimuthal angle θ with a weighted shift factor given by ellipticity angle φ . In detail, the absorption Abs can be calculated by following fitted formula:

$$Abs = a + b \cos(2(\theta + 10)) \quad (1)$$

$$\begin{pmatrix} a \\ b \end{pmatrix} = \begin{pmatrix} a_1 \\ b_1 \end{pmatrix} + \begin{pmatrix} a_2 \\ b_2 \end{pmatrix} (\sin(2\varphi) \cos(2\varphi)) \quad (2)$$

where, a indicates the azimuthal angle independent constant background absorption, b indicates the amplitude of θ -resolved absorption component with a fixed φ . The a_1 and a_2 represent the constant component and the amplitude of the φ -resolved a component as a function of sine, and b_1 and b_2 represent the constant component and the amplitude of the φ -resolved b component as a function of cosine, respectively.

$$\begin{pmatrix} a_1 \\ b_1 \end{pmatrix} \sim \begin{pmatrix} 63.15 \\ 0 \end{pmatrix} \quad (3)$$

$$\begin{pmatrix} a_2 \\ b_2 \end{pmatrix} \sim \begin{pmatrix} 19.70 \\ 13.16 \end{pmatrix} \quad (4)$$

Here, the values in the matrix are extracted from the fitting of simulation results (the details of the extraction process are presented in Supplementary Note 1).....

On page 2 of the revised Supplementary Information.

Supplementary Note S1: State of polarization dependent absorption of the chiral metamaterials

As shown in the main manuscript, the state of polarization dependent absorption can fit well with a cosine function of the azimuthal angle θ with a weighted shift factor given by ellipticity angle φ . In detail, the extraction process of the coefficients (a_1 , a_2 , b_1 , b_2) in the equation is as follows:

Firstly, by fitting the azimuthal angle θ dependent absorption with a fixed ellipticity angle φ using a cosine function, a series of coefficients pairs (a , b) can be obtained. Secondly, the coefficient a as a sine function of ellipticity angle φ is fitted and two coefficients (a_1 and a_2) representing the constant component and the amplitude of the φ -resolved component can be

obtained. Thirdly, the coefficient b as a cosine function of ellipticity angle φ is fitted and two coefficients (b_1 and b_2) representing the constant component and the amplitude of the φ -resolved component can also be obtained.

Comment 4: How did the authors obtain equations 7, 8, 9 on page 12? I could not find the corresponding discussion and derivation. The authors could include the explanation in the method section if it is too lengthy.

Response to reviewer: We thank the reviewer for the constructive and useful suggestion. According to the reviewer's suggestion, we added the derivation of the expressions for calculation of geometrical ellipse parameters in the revised Supplementary Information. The corresponding description has been added in the revised manuscript, also copied below:

Lines 3-5, on page 13 of the revised manuscript.

where C refers to the maximum photovoltage output under LCP or RCP light illumination with a typical incident power (the derivation of these expressions is presented in the Supplementary Note S2).

On pages 3-4 of the revised Supplementary Information.

Supplementary Note S2: Derivation of the expressions for calculation of geometrical ellipse parameters

As we designed, the photovoltage output of each port is polarization dependent. Particularly, the photovoltage outputs of Port 1 and Port 2 are linear and circular polarization dependent, and the photovoltage output of Port 3 is only circular polarization dependent. The general expression of the photovoltages for each part in our designed device can be expressed as:

$$V_{ph} = L_i \sin(2\theta) + C_i \tan(\varphi) \quad (S1)$$

where, $L_i \cdot \sin(2\theta)$ is the linear-polarization-resolved (θ -resolved) photovoltage component, and $C_i \cdot \tan(\varphi)$ is the circular-polarization-resolved (φ -resolved) photovoltage component. No constant background photoresponse is available due to the bipolar responses of our devices. In detail, the expression of each port in our three-ports device can be expressed as:

$$P_1 = (L_1 \sin(2(\theta + 45)) + 0.5C_1 \tan(\varphi)) - (L_0 \sin(2\theta) - 0.5C_0 \tan(\varphi)) \quad (S2)$$

$$P_2 = (L_2 \sin(2(\theta + 135)) + 0.5C_2 \tan(\varphi)) - (L_0 \sin(2\theta) - 0.5C_0 \tan(\varphi)) \quad (S3)$$

$$P_3 = (L_3 \sin(2(\theta)) + C_3 \tan(\varphi)) - (L_0 \sin(2\theta) - C_0 \tan(\varphi)) \quad (S4)$$

Here, the coefficients (L_i and C_i) for each port output can be obtained by calibration with the experimental results. Based on the experimental results as shown in Figure S19, we can

simplify the Eq. S2-4 as:

$$P_1 = (L'_1 \sin(2(\theta - 112.5)) + C'_1 \tan(\varphi)) \quad (\text{S5})$$

$$P_2 = (L'_2 \sin(2(\theta - 67.5)) + C'_2 \tan(\varphi)) \quad (\text{S6})$$

$$P_3 = -C'_3 \tan(\varphi) \quad (\text{S7})$$

According to fitting results with experimental results, the coefficients can meet relationships as: $L''=L_1'=L_2'$, and $C''=C_1'=C_2'=C_3'$. Therefore, the Eq. S5-7 can be further simplified as:

$$P_1 = (L'' \sin(2\theta - 45) + C'' \tan(\varphi)) \quad (\text{S8})$$

$$P_2 = (L'' \cos(2\theta - 45) + C'' \tan(\varphi)) \quad (\text{S9})$$

$$P_3 = -C'' \tan(\varphi) \quad (\text{S10})$$

Therefore, the azimuthal angle θ and ellipticity angle φ can be calculated as:

$$\theta = \frac{1}{2} \left(\tan^{-1} \frac{P_1 + P_3}{P_2 + P_3} + 45 \right) \quad \text{when } P_2 < 0 \quad (\text{S11})$$

$$\theta = \frac{1}{2} \left(\tan^{-1} \frac{P_1 + P_3}{P_2 + P_3} + 45 \right) + 90 \quad \text{when } P_2 > 0 \quad (\text{S12})$$

$$\varphi = \tan^{-1} \frac{P_3}{C} \quad (\text{S13})$$

where only one coefficient (C) is needed to be extracted, which is related to the incident light power.

Comment 5: Figure 4 shows the results regarding the stokes parameter detection. But the plots do not show clearly the measurement accuracy for stokes parameters, for comparison with other techniques presented in the literature. It would be much more straightforward to show the measurement results and extracted errors for stokes parameters, S1, S2, and S3, on these plots. The authors should also add some discussions about the measurement errors and indicate fundamental limitations, e.g., cross-talk, of their method and the corresponding impact on the measurement error if there are any.

Response to reviewer: We thank the reviewer for the constructive and useful suggestion. Following the reviewer's suggestion, we added the polarimetry accuracy analysis of the three-ports device by calculating the average errors of retrieved Stokes parameters in the revised Supplementary Information. The corresponding description and discussion about the measurement errors have also been added in the revised manuscript, as copied below:

Lines 19-27, on page 13 of the revised manuscript.

.....To evaluate the polarimetry accuracy of our three-ports device, the deviation of three Stokes parameters of a set of measurement results are calculated and shown in Supplementary Figure S21. The average measurement errors of S_1 , S_2 , and S_3 are 14.2%, 15.2%, and 5.4%, respectively. The relative higher measurement errors for S_1 and S_2 than that for S_3 can be attributed to the nonimmune photoresponses of Port 1 and Port 2 against the circular polarization. On the other hand, the imperfect fabrication of metamaterials, the imperfect Gaussian distribution of laser beam, the error of input light polarization and so on will also introduce the measurement errors of Stokes parameters.

On page 25 of revised Supplementary Information.

Fig. S21 The polarimetry accuracy of the three-ports device. **a**, The retrieved Stokes parameters with different polarization states. The hollow marks indicate input polarization states, and solid marks indicate measured polarization states. **b**, The calculated average errors of three Stokes parameters (S_1 , S_2 , and S_3).

Comment 6: What is the operating temperature of the device? Is it at room temperature? Would the device work better at a lower temperature? What is the detection speed (time for measuring one polarization state)?

Response to reviewer: The operating temperature of the device is at room temperature. On the other hand, we measured the photoresponse of the device at different temperatures

and added corresponding description in the revised manuscript. As for the detection speed, the photoresponse time of the device is about $76 \mu\text{s}$ corresponding to a -3dB bandwidth of 1.1 kHz . Therefore, the time of measuring one polarization state is also about $76 \mu\text{s}$ if we suppose the photoresponse speed of the device is the slowest bottleneck. Corresponding additional descriptions are also copied below:

Lines 1-8, on page 10 of the revised manuscript.

..... Moreover, the detector exhibits a high responsivity up to 3.6 V W^{-1} , a short response time of $76 \mu\text{s}$ corresponding to a -3dB bandwidth of 1.1 kHz , a low dark noise spectrum down to $35 \text{ nV Hz}^{-1/2}$ corresponding to a noise-equivalent power of $9.7 \text{ nW Hz}^{-1/2}$ and a specific detectivity of $2.5 \times 10^5 \text{ Jones}$, and good response repeatability and stability at room-temperature. Furthermore, the device exhibits a lower photoresponse at lower temperatures (Supplementary Figure S11), which results from the low temperature gradient owing to the efficient heat dissipation or high thermal conductivity at low temperatures.^{44, 45}

On page 16 of revised Supplementary Information.

Fig. S11 Temperature dependent photoresponse. a, The photoresponses at different temperatures. **b,** The temperature-dependent photovoltages with an incident light power of $38 \mu\text{W}$.

Comment 7: The authors used an object on a motorized 2D stage to complete a polarization mapping. This is not the same as what people usually refer to as mid-IR imaging. From a single element polarization detection device to a real polarimetric imaging sensor, one needs to consider the pixel size, the cross-talk between adjacent pixels, and the scalability of the read-out circuit.

Response to reviewer: We thank the reviewer for the comment. Here, in this work, we demonstrated a single-pixel detector concept for retrieval of full-stoke polarization state and

used it for a polarization imaging demonstration. Indeed, for practical polarization real imaging applications, one needs to use a two-dimensional array detector, however, it is beyond the scope of the current work. From a single element polarization detection device to a real polarimetric imaging sensor, one needs to consider the pixel size, the cross-talk between adjacent pixels, and the scalability of the read-out circuit, which we would like to explore for our future studies. To clearly describe this issue, we added corresponding discussion in the revised manuscript, also copied below:

Lines 21-25, on page 14 of the revised manuscript.

.....Therefore, the proposed polarimeter shows great practical application potentials in mid-IR polarimetric imaging, although there are some needful elements needed to be considered for the practical imaging application, including the size of single pixel, the crosstalk between adjacent pixels, the scalability of the read-out circuit, and so on.

REVIEWERS' COMMENTS

Reviewer #1 (Remarks to the Author):

The results seem interesting and are worthy of reference and creativity. I recommend the publication on Nat. Commun. modifications.

Reviewer #2 (Remarks to the Author):

I looked at the changes, they are fine. The manuscript is now acceptable for publication.

Reviewer #3 (Remarks to the Author):

The revised version addressed all the comments in a satisfactory manner. I recommend it for publication with some minor revisions in the language to improve the readability.

For example, on page 14 of the revised manuscript, the author writes:

..... although there are some needful elements needed to be considered for the practical imaging application, including the size of single pixel, the crosstalk between adjacent pixels, the scalability of the read-out circuit, and so on.

"needful elements needed to be considered for...application"... Don't the authors mean simply "necessary elements for...application"?

Similar language improvements can be made in many places of the manuscript for a better reading experience for the readers.

Reply to the Reviewers' Comments

We are grateful to the kind and constructive advice on our manuscript which has been submitted to Nature Communications (ID: NCOMMS-22-14122A. Title: On-chip mid-infrared photothermoelectric detectors for full-Stokes detection). In response to these comments, we have revised the manuscript and the supplementary information accordingly. We believe that we have addressed all the comments raised by the editor and reviewers.

The corresponding amendments addressing the editor and referees' concerns have been marked with yellow background for easy reference in the revised manuscript (Revised Manuscript-with marks). Our replies to the comments by the editor and reviewers and the corresponding changes are listed below.

The following are details of our point-by-point responses to reviewers' comments:

Reviewer #1:

Comments:

The results seem interesting and are worthy of reference and creativity. I recommend the publication on Nat. Commun. modifications.

Response to reviewer: We really thank the reviewer for the constructive suggestions/comments that have greatly helped us to improve the quality and depth of our work.

Reviewer #2:

Comments:

I looked at the changes, they are fine. The manuscript is now acceptable for publication.

Response to reviewer: We appreciate the valuable comments from the reviewer.

Reviewer #3:

Comments:

The revised version addressed all the comments in a satisfactory manner. I recommend it for publication with some minor revisions in the language to improve the readability.

Response to reviewer: We thank the reviewer for the constructive comments.

Comment 1: For example, on page 14 of the revised manuscript, the author writes:

although there are some needful elements needed to be considered for the practical imaging application, including the size of single pixel, the crosstalk between adjacent pixels, the scalability of the read-out circuit, and so on.

"needful elements needed to be considered for...application"... Don't the authors mean simply "necessary elements for...application"?

Similar language improvements can be made in many places of the manuscript for a better reading experience for the readers.

Response to reviewer: We thank the reviewer for the careful reading of our manuscript. We have carefully revised our manuscript to avoid any grammatical or spelling errors. In addition, according to reviewer's suggestion, we have made some revisions in the language to improve the readability, also copied below:

Lines 18-20, on page 5 of the revised manuscript.

.....and a polarization imaging demonstration is presented with the developed device. Our results show a filterless, uncooled, bandgap-independent, devisable wavelength-specific.....

Lines 29-30, on page 6 of the revised manuscript.

.....The temperature distribution in our device is simulated with a consideration of the heat conductance, radiation, and convection.....

Lines 6-10, on page 7 of the revised manuscript.

.....To investigate the photothermal response of the chiral plasmonic metamaterials, full-wave electromagnetic simulations were performed on the Z-shaped Au antenna array. Firstly, structural parameters as indicated in Figure 1b are obtained using global optimization. The optical absorption of chiral metamaterials can be tuned across a broadband mid-IR regime.....

Lines 23-24, on page 9 of the revised manuscript.

.....This indicates that the operation wavelength of our proposed approach is no longer limited by the bandgap of the active material.....

Lines 20-21, on page 10 of the revised manuscript.

.....show good agreements with the calculated results in the configurable polarity transition.....

Lines 29-30, on page 10 of the revised manuscript.

Thanks to the configuration flexibility of metamaterials, the proposed resonant PTE response also enables us to realize circular polarization detection.....

Lines 12-15, on page 14 of the revised manuscript.

For different polarizations of the incident light, all three ports (left three columns in Figure 5b-e) can obtain clearer polarization imaging results and exhibit a typical combining form, which is a one-to-one correspondence to the SoP.....

Lines 22-24, on page 14 of the revised manuscript.

..... For practical imaging applications, the size of single pixel, the crosstalk between adjacent pixels, and the scalability of the read-out circuit, also need to be considered.

Lines 1-2, on page 15 of the revised manuscript.

In summary, we have presented mid-IR polarization-sensitive PTE detectors with several advantages such as filterless, uncooled, bandgap-independent, tailorable operating.....